

# Covariability of dynamics and composition in the Asian monsoon tropopause layer from satellite observations and reanalysis products

Shenglong Zhang[1,2], Jiao Chen[1], Jonathon S. Wright[1], Sean M. Davis[3], Jie Gao[4], Paul Konopka[5], Ninghui Li[1], Mengqian Lu[2], Susann Tegtmeier[6], Xiaolu Yan[7], Guang J. Zhang[8], and Nuanliang Zhu[9]

[1]Department of Earth System Science, Tsinghua University, Beijing, China
[2]Department of Civil and Environmental Engineering, The Hong Kong University of Science and Technology, Hong Kong, China
[3]Earth System Research Laboratory, National Oceanic and Atmospheric Administration, Boulder, Colorado, USA
[4]Laoshan Laboratory, Qingdao, China
[5]Forschungszentrum Jülich (IEK-7: Stratosphere), Jülich, Germany
[6]Department of Physics and Engineering Physics, University of Saskatchewan, Saskatoon, Canada
[7]Institute of Tibetan Plateau Meteorology, Chinese Academy of Meteorological Sciences, Beijing, China
[8]Scripps Institution of Oceanography, La Jolla, California, USA
[9]Department of the Geophysical Sciences, University of Chicago, Chicago, Illinois, USA

**Correspondence:** Jonathon S. Wright (jswright@tsinghua.edu.cn)

**Abstract.** We describe three leading modes of deseasonalized water vapor variability in the tropopause layer (147–68 hPa) above the Asian summer monsoon (ASM) based on Aura Microwave Limb Sounder (MLS) satellite observations and five meteorological and composition-focused reanalyses. The first mode, which describes regional-scale moist or dry anomalies on interannual scales, is separated into a linear trend and detrended interannual variability. Although the reanalysis products all

show an increasing trend in tropopause-layer water vapor over 2005–2021, the spatial pattern and sign of the trend disagree between Aura MLS and the reanalyses. The regional water vapor budget indicates that the reanalysis trend originates outside the monsoon region, beyond the domain of our analysis. Interannual variability is otherwise consistent, arising mainly from the pre-monsoon influence of the quasi-biennial oscillation. The second mode features anomalies arcing from the southwestern to northeastern quadrants of the anticyclone coupled with weaker opposing anomalies in the southeast, while the third mode

features a horizontal dipole oriented east-to-west. These two modes often vary in quadrature due to the influences of quasi-biweekly waves on deep convective activity, but also appear independently when other modes of convective variability manifest in similar centers of action. Although questions remain regarding the linear trend, mean biases, and the weak and possibly adverse influence of data assimilation, the consistency between Aura MLS and reanalysis-derived modes of variability in UTLS water vapor in this region shows that atmospheric reanalyses are increasingly able to capture the processes controlling

water vapor near the tropopause.

## 1 Introduction

Although water vapor is much less abundant in the stratosphere than in the troposphere, changes in the distribution and amount of stratospheric humidity have important influences on stratospheric chemistry and the global radiative balance (Forster and



Shine, 1999; Solomon et al., 2010; Riese et al., 2012; Dessler et al., 2013; Maycock et al., 2013; Banerjee et al., 2019). Strong
deep convection and the upper-level circulation associated with the Asian summer monsoon (ASM) produce clear signals in
lower stratospheric water vapor and other constituents during boreal summer (e.g. Milz et al., 2005; Park et al., 2007; Santee
et al., 2017; Kumar and Ratnam, 2021), and many studies have emphasized the importance of transport through this region to
humidity and composition in the lower stratosphere (e.g. Dethof et al., 1999; Bannister et al., 2004; Gettelman et al., 2004;
Fueglistaler et al., 2005; Wright et al., 2011; Ploeger et al., 2013; Pan et al., 2016; Nützel et al., 2019; Yan et al., 2019).
Lower stratospheric water vapor (LSWV) above the ASM is primarily controlled by temperatures near the tropopause (James
et al., 2008; Randel et al., 2015). However, very deep convective storms can substantially raise LSWV concentrations lo-
cally (Ueyama et al., 2018; Lee et al., 2019), and may also be a source of variations in water vapor above the tropopause at
longer time scales (Brunamonti et al., 2018, 2019; Singh et al., 2021). Despite much progress in recent years, uncertainties
remain regarding the relative influences and interplays among convective transport, the large-scale circulation, and thermody-
namic structure near the tropopause in controlling humidity and composition in the upper troposphere and lower stratosphere
(UTLS) above the ASM.

The seasonal cycle and interannual variability of LSWV in the tropics are intrinsically linked to variations in temperature at
and around the tropical tropopause (Mote et al., 1996; Randel et al., 2006; Liu et al., 2010; Schoeberl et al., 2012), with the
coldest temperatures encountered during transit effectively determining how much water vapor enters the stratosphere (Gettel-
man et al., 2002; Fueglistaler et al., 2005; Randel and Jensen, 2013). Focusing on the 100 hPa isobaric surface, Randel et al.
(2015) reported that variations in LSWV above the Asian summer monsoon (ASM) are also controlled mainly by variations in
temperatures and the large-scale circulation. The most influential circulation system in this sense is the upper-level monsoon
anticyclone, for which the coldest temperatures are concentrated along the southern flank over the Bay of Bengal and northern
India (e.g. Wright et al., 2011; Zhang et al., 2016). The smooth boundaries and distinct shape of the climatological anticyclone
mask vigorous variations on a wide range of timescales (Garny and Randel, 2013; Ploeger et al., 2015; Pan et al., 2016; Nützel
et al., 2016; Yan et al., 2018; Ren et al., 2019; Siu and Bowman, 2020) that influence the extent to which air in the monsoon
UTLS is confined and isolated from air outside the anticyclone boundaries (Ploeger et al., 2013; Nützel et al., 2019; Legras
and Bucci, 2020; Honomichl and Pan, 2020; Kumar and Ratnam, 2021). The origins and full impacts of this variability are not
yet well constrained. Moreover, studies examining water vapor and other tracers at and above the tropopause within the ASM
anticyclone have reached diverse conclusions on the importance of direct injection by overshooting convection, ranging from
marginal (e.g. James et al., 2008; Randel et al., 2015; von Hobe et al., 2021; Konopka et al., 2023) to substantial (15% or more;
e.g. Ueyama et al., 2018; Wang et al., 2019). The seemingly contradictory nature of these conclusions is rooted in the difficulty
of observing and modeling convective influences near the tropopause. However, despite some quantitative discrepancies, re-
cent reanalyses largely reproduce observed seasonal-mean anomalies in water vapor near the tropopause in this region (Wright
et al., 2025).

Variations in ozone and carbon monoxide (CO) have close associations with water vapor in the UTLS and provide com-
plementary information on the processes affecting water vapor in the monsoon tropopause layer (Pan et al., 2016; Gottschaldt
et al., 2018; von Hobe et al., 2021). Stratospheric ozone influences dynamical and thermodynamic conditions through its in-



teractions with shortwave and longwave radiation, and UTLS ozone can influence stratospheric water vapor by modulating
temperatures near the tropical tropopause (Ming et al., 2017; Xia et al., 2018) and altering high cloud distributions through
temperature effects on relative humidity (e.g. Xia et al., 2016, 2018). Seasonal dilution in the UTLS above the monsoon results
in a local minimum that has been dubbed the 'ozone valley' (Bian et al., 2011). Ozone also serves as a stratospheric tracer, with
convective dilution in the core of the anticyclone (Bian et al., 2011) flanked by higher concentrations due to in-mixing from the
extratropical lower stratosphere (Konopka et al., 2009, 2010). CO, as a predominantly tropospheric tracer, is a valuable marker
of convective uplift of polluted air to the UTLS (Huang et al., 2014; Pan et al., 2016; Santee et al., 2017), and is often used as
a proxy for local enhancements in aerosol concentrations or as a transport tracer to track the fate of polluted boundary-layer
air (e.g. Pan et al., 2022).

In this study, we investigate the spatiotemporal signatures of variability in UTLS water vapor above the ASM at both inter-
annual and subseasonal time scales and link these signatures to covariations in ozone, CO, and dynamical and thermodynamic
fields from reanalyses. We conduct this analysis using satellite data from the Aura Microwave Limb Sounder (MLS) and prod-
ucts from five recent atmospheric reanalyses (see Sect. 2). This study aims to provide further insight into the mechanisms
governing variations in water vapor and other trace gases in the UTLS above the ASM as represented in current atmospheric
reanalysis systems. We pay particular attention to characteristic patterns of covariability among convective activity, circulation
patterns, and trace gas concentrations. Much of the foundation for this work has been laid in Chapter 8 (Tropical Tropopause
Layer) of the Stratosphere-troposphere Process and their Role in Climate (SPARC) Reanalysis Intercomparison Project (S-RIP)
Final Report (Tegtmeier et al., 2022, and references therein), along with our recent examination of reanalysis representations
of climatological mean composition in this region (Wright et al., 2025).

The remainder of this paper is organized as follows. In Sect. 2, we introduce the data and methods used in the analysis. In
Sect. 3, we use principal component analysis to identify and analyze the leading modes of variability in deseasonalized water
vapor anomalies above the ASM considering both horizontal and vertical structure at time scales 5 days and longer, and explore
the mechanisms behind these leading modes of variability. To support this central section, we provide an online supplement
in which the main elements of the analysis are repeated using different reanalysis products. In Sect. 4, we compare our results
with those of other recent studies, before concluding with a brief summary in Sect. 5.

## 2  Data and method

### 2.1  Data

We use Microwave Limb Sounder (MLS; Waters et al., 2006) version 5 (v5) gridded retrievals of water vapor (Lambert et al.,
2021), ozone (Schwartz et al., 2021a), and carbon monoxide (Schwartz et al., 2021b) from 2005–2021 covering the ASM
region (30°E–130°E and 15°N–45°N). Daily gridded data are averaged into pentads (discrete 5-day periods) starting from
1st May and ending on 2nd October, totaling 31 pentads per year (155 days). We focus on humidity in the monsoon tropopause
layer between 147 hPa and 68 hPa (approximately 370–440 K potential temperature).



In addition to MLS observations, we use model-level and pressure-level outputs from five atmospheric reanalyses. These include the European Centre for Medium-Range Weather Forecasts (ECMWF) Fifth Reanalysis of the Atmosphere (ERA5; Hersbach et al., 2020), the Japanese Reanalysis for Three Quarters of a Century (JRA-3Q; Kosaka et al., 2024), the Modern-Era Retrospective Analysis for Research and Applications, version 2 (MERRA-2; Gelaro et al., 2017), the MERRA-2 Stratospheric
Composition Reanalysis of Aura Microwave Limb Sounder (M2-SCREAM; Wargan et al., 2023), and the Copernicus Atmosphere Monitoring Service (CAMS) reanalysis of atmospheric composition (Inness et al., 2019). Among these, ERA5, JRA-3Q, and MERRA-2 are focused primarily on meteorological fields, while M2-SCREAM and CAMS are focused more on atmospheric composition. Accordingly, M2-SCREAM and CAMS feature more sophisticated chemistry and additional assimilated variables relative to the corresponding meteorological reanalyses MERRA-2 and ERA5, while the meteorological reanalyses
provide a richer set of dynamical and thermodynamic diagnostics for mechanistic analysis. Key details of these five reanalysis systems, including model resolutions and a catalogue of how assimilated observations affect water vapor, ozone, and CO at these altitudes, have been provided by our companion paper (Wright et al., 2025).

We use these reanalysis products to support a deeper investigation into recurrent patterns of variability in LSWV, both to compare the consistency of internal variations to those observed by Aura MLS and to evaluate the potential mechanisms behind
the observed variations. Three metrics are adopted to evaluate the effects of deep convection and thermodynamic conditions during transport. First, cold point tropopause (CPT) temperatures are identified using instantaneous model-level temperature profiles from ERA5 and MERRA-2 and pressure-level temperature profiles from JRA-3Q as described by Tegtmeier et al. (2020). Second, all-sky outgoing longwave radiation (OLR) fields from ERA5, MERRA-2, and JRA-3Q are used to indicate the prevalence of cold high clouds, in line with previous studies on variations in the overall frequency and spatial extent of deep
convection in this region (e.g. Randel et al., 2015). Variations in very deep convection are also evaluated using non-radiative diabatic heating, which primarily indicates the strength of convective ascent in the reanalysis model. Anomalous non-radiative heating at these levels is dominated by variations in the deepest simulated convective towers. Although such convection is rare, heating rates associated with these events are still approximately 1–2 orders of magnitude larger than contributions from parameterized shear-flow turbulence close to the tropopause (Wright and Fueglistaler, 2013; Tegtmeier et al., 2022). These
heating rates are model products and are thus not directly constrained by data assimilation; however, they exert strong influences on composition and humidity in the reanalysis tropopause layer. Model-generated heating rates are only available for ERA5, MERRA-2, and JRA-3Q. Given the overlap in forecast models and, in the latter case, the use of the 'replay' technique, heating rates for ERA5 may be taken as roughly representative of those in CAMS, and similarly for MERRA-2 and M2-SCREAM.

Direct data assimilation constraints on humidity are weak at these levels (Wright et al., 2025). Different reanalyses deal
with this lack of strong constraints in different ways. For example, the ECMWF reanalyses, ERA5 and CAMS, suppress direct assimilation increments in humidity at these levels by setting background error covariances to zero (although assimilation increments in winds and temperatures still influence the evolution of humidity fields at these altitudes). MERRA-2 only assimilates radiosonde humidities at pressures greater than 300 hPa, whereas JRA-3Q restricts radiosonde humidity assimilation to temperatures greater than $-40°$C. Moreover, although satellite radiances can affect humidity at all altitudes in these two
systems, small absolute concentrations mean that satellite radiances provide little information about water vapor at these al-





titudes. Because of these weak direct constraints and the deep vertical weighting functions that apply for some assimilated radiances, analysis increments targeted at the troposphere may produce increments at and above the tropopause as well. This problem produced large biases in stratospheric humidity in JRA-55 (Davis et al., 2017; Kosaka et al., 2024), which persist in a less pronounced form in JRA-3Q (Wright et al., 2025). MERRA-2 sidesteps the issue by relaxing stratospheric humidity

to a monthly-mean zonal-mean climatology (Gelaro et al., 2017); however, this approach necessarily suppresses water vapor anomalies at these altitudes, especially those on longer timescales (Davis et al., 2017). M2-SCREAM is unique among the reanalyses considered here in that it directly assimilates Aura MLS retrievals of water vapor using the Goddard Earth Observing System (GEOS) Constituent Data Assimilation System (CoDAS; Wargan et al., 2023). Accordingly, comparisons between M2-SCREAM and Aura MLS are not independent.

We also use several reanalysis-based variables to describe the dynamical state of the monsoon tropopause layer. First, we use Montgomery streamfunction (MSF $= c_p T + \Phi$, where $c_p$ is the specific heat of dry air at constant pressure, $T$ is temperature, and $\Phi$ is geopotential) on the 395 K isentropic surface to track the position and strength of the anticyclone (Santee et al., 2017; Manney et al., 2021). MSF, which is functionally identical to dry static energy, defines the geostrophic wind on an isentropic surface and is thus analogous to geopotential in pressure coordinates. The 395 K isentropic surface is typically located

between 100 hPa and 83 hPa in the core ASM region south of the subtropical westerly jet. Second, we use horizontal winds, temperatures, ozone mass mixing ratios, and CO mass mixing ratios from each reanalysis to evaluate the dynamical transport environment within the ASM UTLS. Winds and temperatures are used to describe the structure of the anticyclone, while ozone and CO mass mixing ratios are used to qualitatively indicate anomalous in-mixing of stratospheric air and convectively detrained air, respectively (e.g. Pan et al., 2022). Among the reanalyses, only CAMS and MERRA-2 provide estimates of CO and

only CAMS includes any CO-related data assimilation (Inness et al., 2019; Wright et al., 2025).Finally, we use specific humidity tendencies due to parameterized physics, data assimilation, and dynamical transport from ERA5, JRA-3Q, and MERRA-2. These three meteorological reanalyses are used because they provide enough of the terms to close the vertically-resolved water vapor budget. As with diabatic heating, tendencies from ERA5 may be considered roughly representative of those for CAMS; however, tendencies from MERRA-2 are not representative of those for M2-SCREAM because these two systems treat strato-

spheric water vapor very differently as noted above: MERRA-2 nudges to a climatology, while M2-SCREAM assimilates Aura MLS.

## 2.2 Methodology

Motivated by the analysis of Randel et al. (2015), who investigated the mechanisms behind water vapor variability at 100 hPa, we evaluate coupled horizontal and vertical variations of LSWV above the ASM in the 147–68 hPa layer, including observed

and reanalysis water vapor mixing ratios. This layer, which contains the CPT, is a transitional zone where air detrained from convection spirals upward into the lower stratosphere (e.g. Vogel et al., 2019). We integrate variables in pressure from $p_{\text{bot}}$ (147 hPa) to $p_{\text{top}}$ (68 hPa) and refer to the integrated variable as partial-column water vapor (PCWV) in units of mass per area:



$$\text{PCWV} = -\frac{1}{g} \int\limits_{p_{\text{bot}}}^{p_{\text{top}}} q \, dp, \tag{1}$$

where $g$ is the gravitational acceleration, $q$ is specific humidity, and $dp$ denotes the pressure thickness of each layer between
$p_{\text{top}}$ (68 hPa) and $p_{\text{bot}}$ (147 hPa). Specific humidity from each reanalysis is interpolated from model levels (or pressure levels, in the case of JRA-3Q) to the MLS pressure levels, but no weighting functions are applied. Partial-column ozone (PCO3) and partial-column CO (PCCO) are calculated similarly from ozone and CO mass mixing ratios.

The vertically-resolved water vapor budget is used to diagnose the mechanisms behind UTLS water vapor variability in the meteorological reanalyses:

$$\frac{\partial q}{\partial t} + \nabla \cdot (\mathbf{V}q) + \frac{\partial(\omega q)}{\partial p} = S_{\text{phy}} + S_{\text{ana}} + S_{\text{res}} \tag{2}$$

where $p$ is pressure, $\mathbf{V}$ is the vector horizontal wind on an isobaric surface, and $\omega$ is the pressure vertical velocity. The first term on the left-hand side is the time rate of change, which is computed from assimilated water vapor fields at consecutive time steps. The second and third terms on the left-hand side are the horizontal and vertical moisture flux divergence, which are also computed from assimilated fields and referenced as the 'dynamics' terms below. The terms on the right-hand side represent local sources or sinks. $S_{\text{phy}}$ represents net sources and sinks due to parameterized physical processes in the forecast model, $S_{\text{ana}}$ is the source or sink due to data assimilation, and $S_{\text{res}}$ is the residual. The residual term collects high-frequency and/or small-scale covariance transports that are resolved by the reanalysis model but not by our calculation of moisture flux divergence (Wright et al., 2025). A T42 spectral filter is applied to all budget terms to reduce noise in the spatial patterns. This filter simplifies the visualizations and does not change the results in any significant way.

For subsequent analyses, we average all variables into pentads (discrete 5-day means) starting on 1 May and ending on 2 October, so that the first pentad of each warm season is centered on 3 May and the last is centered on 30 September. We calculate deseasonalized anomalies in each grid cell by subtracting the mean for the corresponding pentad over 2005–2021. We then apply empirical orthogonal function (EOF) analysis to the deseasonalized water vapor anomalies. The EOF analysis considers both horizontal and vertical variations at pentad resolution. Input data are weighted by area in the horizontal dimension, but are weighted equally in the vertical dimension. We retain the three leading principal components (PCs) as indicated by Aura MLS. Eigenvalues associated with these three modes are mutually independent based on the North test (North et al., 1982). Separate EOF analyses are then applied to water vapor volume mixing ratio anomalies from reanalysis products interpolated to the Aura MLS levels and downscaled to the MLS horizontal grid. The PCWV anomalies from each dataset are then regressed onto the corresponding PCs for intercomparison and evaluation of variability as represented by the reanalyses. Some PCs based on reanalysis fields must be reordered for consistency with the Aura MLS results; such cases are noted in the figures and related text. Deseasonalized anomalies in reanalysis fields are then regressed onto the PCs calculated from Aura MLS for a more detailed analysis of the underlying mechanisms. This two-pronged approach provides more stringent tests of the reanalysis products because it evaluates both the extent to which each reanalysis can capture observed spatial patterns





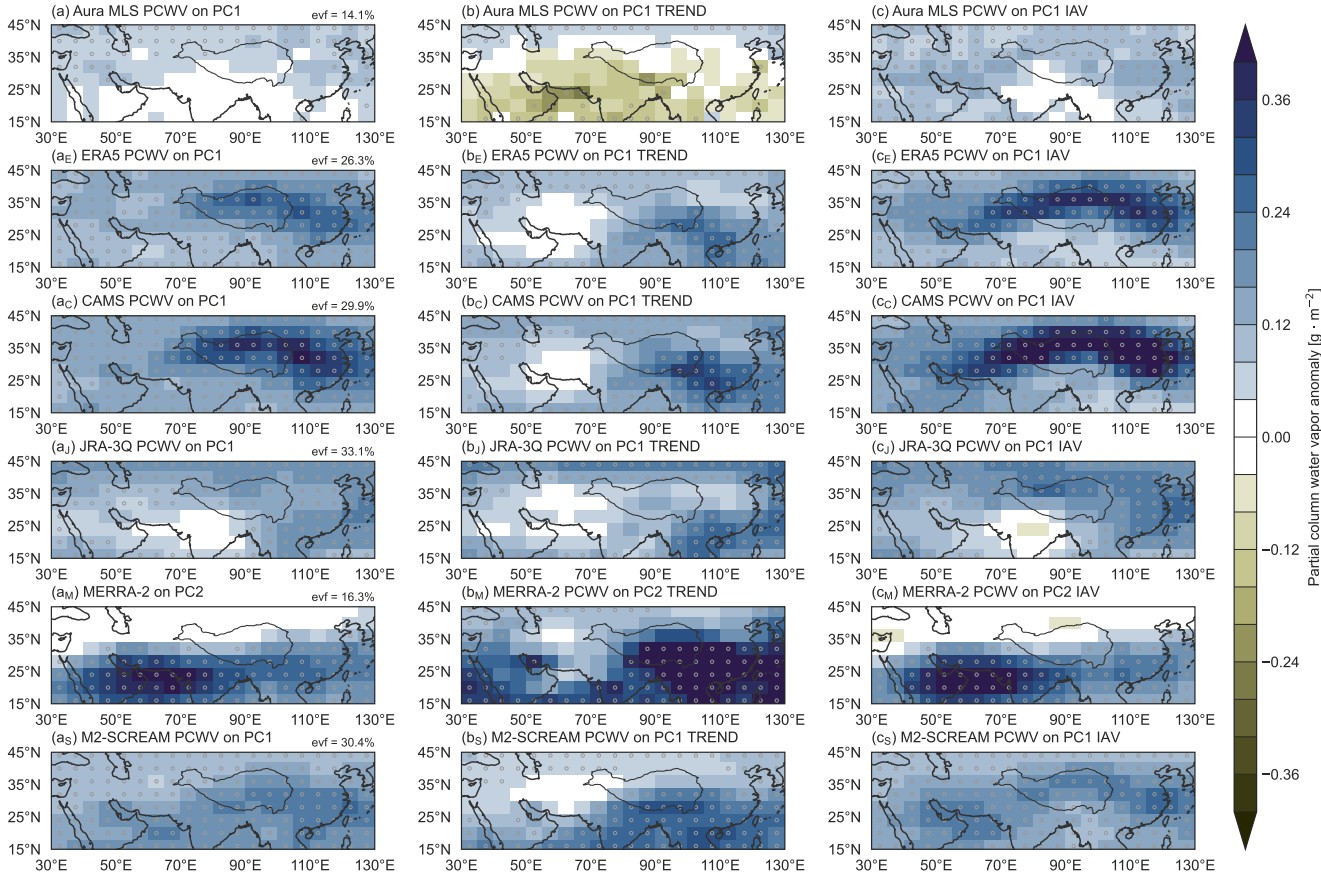

**Figure 1.** Deseasonalized partial-column water vapor (PCWV) anomalies (Eq. 1) regressed onto the (a) first principal component (PC1) from Aura MLS and its (b) trend (PC1$_\mathrm{TREND}$) and (c) interannual variability (PC1$_\mathrm{IAV}$) components. Corresponding distributions are shown for (a$_\mathrm{E}$)–(c$_\mathrm{E}$) ERA5 PCWV on PC1 from ERA5, (a$_\mathrm{C}$)–(c$_\mathrm{C}$) CAMS PCWV on PC1 from CAMS, (a$_\mathrm{J}$)–(c$_\mathrm{J}$) JRA-3Q PCWV on PC1 from JRA-3Q; (a$_\mathrm{M}$)–(c$_\mathrm{M}$) MERRA-2 PCWV on the second principal component (PC2) from MERRA-2; and (a$_\mathrm{S}$)–(c$_\mathrm{S}$) M2-SCREAM PCWV on PC1 from M2-SCREAM. Principal components (Fig. 3a–c) are based on EOF analysis of vertical and horizontal variations in water vapor for the five Aura MLS pressure levels within 68 hPa–147 hPa, 30°E–130°E, and 15°N–45°N (Sect. 2.2). The location of the Tibetan Plateau is marked by a dark grey contour and stippling indicates that the regression is significant at the 95% confidence level based on Student's $t$ test. The fraction of total variance explained by each mode is listed at the upper right of panels in column (a).

185  of variability in LSWV (separate PCs) and the extent to which each reanalysis reproduces anomalies in the ASM and UTLS transport environment associated with a given mode (Aura MLS PCs). Combining the second step with the reanalysis-based water vapor budget expressed in eq. (2) allows us to further assess process-level consistency across the reanalyses.



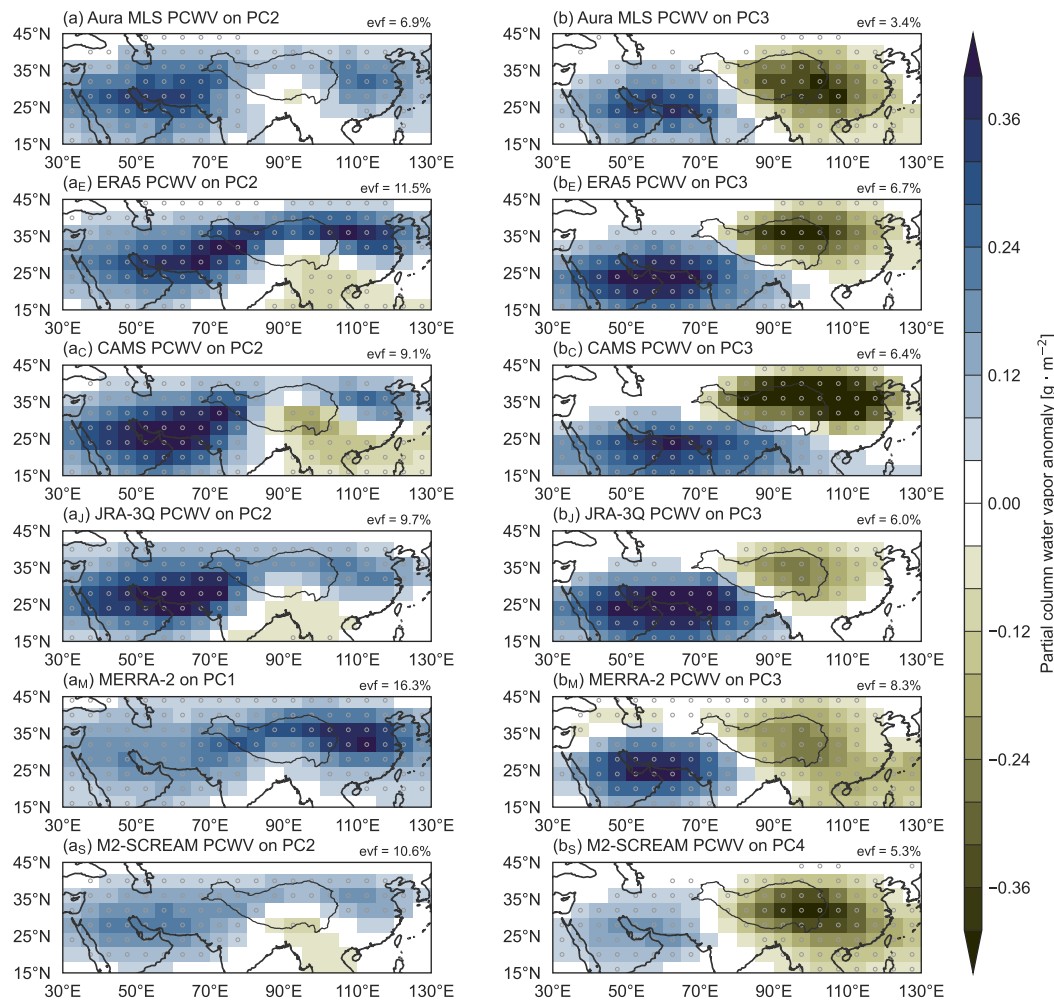

**Figure 2.** As in Fig. 1, but for deseasonalized partial-column water vapor (PCWV) anomalies (Eq. 1) regressed onto the (a) second and (b) third principal components (PCs) from Aura MLS; the ($a_E$) second and ($b_E$) third PCs from ERA5; the ($a_C$) second and ($b_C$) third PCs from CAMS; the ($a_J$) second and ($b_J$) third PCs from JRA-3Q; the ($a_M$) first and ($b_M$) third PCs from MERRA-2; the ($a_S$) second and ($b_S$) fourth PCs from M2-SCREAM. Dark grey contours mark the location of the Tibetan Plateau, with stippling indicating significance at the 95% confidence level based on Student's $t$ test. The fraction of total variance explained by each mode is listed at the upper right of panels.





## 3 Characteristics and mechanisms of variability

Figures 1 and 2 show deseasonalized PCWV anomalies regressed onto leading principal components (PCs) of vertical and hor-
izontal variability in tropopause layer water vapor volume mixing ratios from Aura MLS, ERA5, CAMS, JRA-3Q, MERRA-2,
and M2-SCREAM. The vertical structure of anomalies associated with each mode is discussed below. PC time series for all
datasets are provided in Fig. 3. EOF results for reanalysis products are based on water vapor fields regridded to match the
Aura MLS $2.5° \times 2.5°$ grid and pressure levels, but without applying the Aura MLS vertical weighting functions. The input data
cover the ASM domain (30°E–130°E and 15°N–45°N) on the 147 hPa, 121 hPa, 100 hPa, 83 hPa, and 68 hPa isobaric levels at
pentad temporal resolution.

We retain the three leading PCs from Aura MLS, which are mutually distinct at the 95% confidence level according to
the North test (North et al., 1982) and collectively account for 24% of the variance in Aura MLS. The corresponding spatial
patterns are identifiable in all reanalyses within the first four EOF modes (significant at 90% or greater) and account for 43% of
the water vapor variance in ERA5, 45% in CAMS, 48% in JRA-3Q, 33% in MERRA-2, and 44% in M2-SCREAM. The first
principal component (PC1; column (a) of Fig. 1) accounts for about one sixth to one third of the variance in reanalysis water
vapor in the ASM tropopause layer, with the smallest fraction of explained variance in MERRA-2 (16%). The spatial pattern
based on MERRA-2, with maximum variance in the southwestern quadrant of the anticyclone, is also considerably different
from that identified in other reanalyses. MERRA-2 is a special case because it nudges stratospheric water vapor to a repeating
annual cycle (Gelaro et al., 2017; Avery et al., 2017). Because the leading mode varies primarily on interannual and longer
time scales (Fig. 3a),the nudging applied in MERRA-2 lower stratosphere damps this mode in the upper part of our analysis
domain.

Having no prior expectation of a substantial trend in water vapor in this region over this period, we did not detrend the input
data. However, it is clear from Fig. 3a that PC1 comprises two main types of variability, a linear trend (PC1$_{TREND}$; Fig. 3b)
and interannual variability (PC1$_{IAV}$; Fig. 3c), both of which project onto regional-scale wet or dry anomalies (Fig. 1). We
define the trend component to have a single value each year and a constant increase from year to year (Fig. 3b). The spatial
patterns associated with the trend component (Fig. 1b) represent the main discrepancy between water vapor variability over
the ASM region as observed by Aura MLS versus that represented by the reanalyses during our analysis period. The slopes
of linear trends in all of the PC1 time series are positive. However, whereas Aura MLS regressed onto the PC1$_{TREND}$ time
series indicates a drying trend over 2005–2021 centered in the southwest, the reanalyses show moistening centered in the
southeast. This difference partially explains the larger signal and explained variance in PC1 based on M2-SCREAM, JRA-3Q,
CAMS, and ERA5 relative to Aura MLS. We use version 5 (v5) of Aura MLS, in which the retrieval algorithm has been
designed to mitigate a slow drift toward wet biases in the stratosphere in v4 (Livesey et al., 2021, 2022). M2-SCREAM, which
assimilated Aura MLS v4 throughout our analysis period, provides an indirect indication of the difference between v4 and
v5 in this region. Consistency between M2-SCREAM and the other reanalysis products, which did not assimilate Aura MLS,
thus raises the question: could changes to the retrieval algorithm aimed at removing the spurious drift have also removed real
regional trends? However, trends based on reanalysis products may not be physically meaningful and must be interpreted with

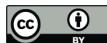



**Figure 3.** Principal component time series for the spatial patterns shown in Figs. 1 and 2 based on Aura MLS (dark grey), ERA5 (dark blue), CAMS (light blue), JRA-3Q (purple), MERRA-2 (dark red), and M2-SCREAM (light red). Time series are shown for (a) the full first principal component (PC1) and its (b) trend (PC1$_{\text{TREND}}$) and (c) interannual variability (PC1$_{\text{IAV}}$) components; (d) the second principal component (PC2); and (e) the third principal component (PC3). Principal components are based on EOF analysis of vertical and horizontal variations in water vapor for the five Aura MLS pressure levels within 68 hPa–147 hPa, 30°E–130°E, and 15°N–45°N (Sect. 2.2). Correlations between MLS-based PCs and those based on MERRA-2 (dark red), M2-SCREAM (light red), JRA-3Q (purple), CAMS (light blue), and ERA5 (dark blue) are listed from right to left along the tops of panels (a), (c), (d), and (e).





caution (e.g. SPARC, 2022), especially over periods as short as this one, and recent work has shown that trends in stratospheric water vapor are difficult to distinguish from multi-decadal variability (Tao et al., 2023). Although the remarkable consistency across reanalysis products gives cause for careful consideration, we remain skeptical of the reanalysis-based trends shown in
column (b) of Fig. 1. Further details are provided in section 3.1.

The largest PCWV anomalies associated with the interannual mode ($PC1_{IAV}$) are located over East Asia and the area southwest of the Tibetan Plateau, with smaller anomalies over South Asia in the convective core of the monsoon (Fig. 1, right column). This spatial structure implies that interannual variability is stronger in the northeastern and southwestern sectors of the anticyclone (i.e. the eddy shedding zones; Popovic and Plumb, 2001; Honomichl and Pan, 2020; Siu and Bowman, 2020) and
weak above and immediately downstream of the strongest convection. With the exception of MERRA-2 as discussed above, correlations between reanalysis-based $PC1_{IAV}$ and that based on Aura MLS are high, with values ranging from 0.72 (CAMS) to 0.93 (M2-SCREAM) indicating that the other four reanalyses reproduce variations in this mode well. The phase of the interannual mode is typically set at the beginning of the monsoon season (Fig. 3c), suggesting the influence of external factors. In most years, the phase is then maintained through the monsoon season, suggesting either steady in-mixing of the anomaly from
outside the anticyclone, maintenance by local feedbacks, strong confinement, or some combination of these. Likely candidates for sources of interannual variability outside the monsoon region include the El Niño–Southern Oscillation (ENSO; Yan et al., 2018; Garfinkel et al., 2021) and the Quasi-Biennial Oscillation (QBO; Ziskin Ziv et al., 2022; Peña Ortiz et al., 2024). We evaluate these possibilities in Sect. 3.2.

The second principal component (PC2; column (a) of Fig. 2) shows moistening or drying centered in the southwestern
quadrant of the anticyclone, with the largest signal centered over the Persian Gulf. This mode explains 7–11% of the variance in datasets other than MERRA-2, for which this mode corresponds to PC1 and explains 16% of the variance (Fig. $2a_M$). Correlations between PC2 from Aura MLS and the corresponding modes in each reanalysis range from 0.55 in MERRA-2 to 0.93 in M2-SCREAM (Fig. 2i), with JRA-3Q ($r = 0.78$), CAMS ($r = 0.77$), and ERA5 ($r = 0.83$) all showing high correlations. Although this mode exhibits some variability at longer timescales, its fluctuations are mainly subseasonal.

The third principal component (PC3), which consists of a southwest–northeast dipole with opposing centers of water vapor anomalies over the Persian Gulf and the eastern Tibetan Plateau, explains slightly more than 3% of the variance in Aura MLS (Fig. 2b). This mode corresponds to the fourth mode in M2-SCREAM (5% of the variance; Fig. 2), and the third mode in all other reanalyses (6–8%). Variations in the corresponding PCs are mainly subseasonal and again show strong correlations with Aura MLS PC3, ranging from 0.59 in MERRA-2 to 0.80 in M2-SCREAM. Spatial patterns show good correspondence
overall, although the southwest-to-northeast tilt of the dipole is sharper in CAMS and ERA5 in comparison to the more east–west orientation in Aura MLS, MERRA-2, and M2-SCREAM. The dipole in JRA-3Q is broadly similar to the latter three but with a stronger signal on the equatorward side of the southwest pole.

Figure 4 shows diabatic heating anomalies along the 25°N transect and zonal mean diabatic heating anomalies averaged across the ASM domain (105°E–130°E). Composite mean differences between the moist (PC ≥ 1) and dry (PC ≤ −1) phases
highlight changes in the intensity and locations of monsoon convection. Here, the 'moist' phase is defined by the presence of positive convective heating and PCWV anomalies in the southwestern quadrant of the anticyclone, as this heating center





**Figure 4.** Composite east–west and north–south transects of diabatic heating anomalies in ERA5 conditioned on (a)–(b) the detrended first principal component (PC1$_{\text{IAV}}$), (c)–(d) the second principal component (PC2), and (e)–(f) the third principal component (PC3) of Aura MLS observations of water vapor variability in the monsoon tropopause layer. Composites are calculated as mean differences between pentads with PC $\geq 1$ and pentads with PC $\leq -1$. Diabatic heating anomalies along the east–west transect (a, c, e) are evaluated along 25°N, while anomalies along the north–south transect (b, d, f) are zonally averaged over 105°E–130°E. Zonal-mean isentropic surfaces are shown in (b) for the moist phase (solid) and dry phase (dotted) of PC1$_{\text{IAV}}$. Composite differences in meridional (c,e) and zonal (d,f) wind for PC2 and PC3 are shown as black contours at contour intervals of $1\,\text{m s}^{-1}$ starting from $\pm 1\,\text{m s}^{-1}$ (solid for southerly and westerly anomalies; dashed for northerly and easterly anomalies).




is present in all three modes and all three reanalyses (Figs. S1–S3 in the online supplement). The moist phase of PC1$_{IAV}$ is associated with enhanced deep convective heating near 70°E (near the Indus River delta), 90°E (over Bangladesh), and 130°E (western North Pacific) in ERA5 (Fig. 4a), with regions of reduced heating in between. These variations largely offset in the

zonal mean, resulting in weak differences in zonally averaged heating (Fig. 4b). Comparison with results for MERRA-2 and JRA-3Q (Fig. S1) shows that increases in convective heating west of 70°E and near 130°E are robust among the three datasets but differences between 70°E and 130°E are not. The largest discrepancies are over Bangladesh and the Bay of Bengal (around 85°E–95°E), where ERA5 shows stronger convection in the moist phase (Fig. S1a), MERRA-2 shows stronger convection in the dry phase (Fig. S1b), and JRA-3Q shows little difference between the phases. Figure 4b also shows uplift of the isentropic

surfaces along the northern flank of the anticyclone in the moist phase relative to the dry phase, indicative of weaker UTLS baroclinicity, and a weaker tropical easterly jet equatorward of 25°N.

The moist phase of PC2 (Fig. 4c–d) is associated with enhanced convective heating in the west and reduced convective heating in the east along the 25°N transect, with zonal-mean heating enhanced south of this transect (15°N–20°N) and reduced to the north (25°N–30°N). These differences are consistent across all three reanalyses (Fig. S2), with the strongest signal in

MERRA-2 and the weakest signal in JRA-3Q. Differences in convective heating along 25°N are located directly below strong differences in the meridional winds (Fig. 4c), with southerly anomalies over weakened convection and northerly anomalies over strengthened convection indicating an upper-level cyclonic anomaly during moist phases and an anticyclonic anomaly during dry phases. The zonal wind response (Fig. 4d) suggests a northward shift of the upper-level anticyclone.

Positive PC3 (Fig. 4e–f) is linked to enhanced deep convective heating over a broad region from 60°E to 100°E along the

25°N transect, with stronger zonal mean heating in the upper troposphere at all latitudes south of 30°N. Figure 4e also shows anomalies in meridional winds along the 25°N transect, which suggest a wave train along the subtropical westerly jet with an cyclonic center around 110°E and a anticyclonic center around 55°E. In the zonal mean, both the subtropical westerlies and the tropical easterlies are weaker in the positive phase and stronger in the negative phase (Fig. 4f). All three reanalyses indicate increases in the depth of convection along the southern flank of the anticyclone during the moist phase relative to the dry phase

(Fig. S3). Distributions of anomalous heating and winds are likewise broadly consistent.

The following sections present detailed analyses of the interannual (PC1) and subseasonal (PC2 and PC3) modes, including changes in composition (based on CAMS) and the dynamical and thermodynamic environment (based on ERA5). Results for other reanalyses and inter-reanalysis differences are provided in the supplement, with brief descriptions provided in the main text. For comparability, the following results are all based on principal components calculated based on Aura MLS.

## 3.1 The trend component

The largest difference between Aura MLS and the reanalysis products in our results is in the linear trend component of PC1. Whereas Aura MLS suggests a drying trend during 2005—2021 centered in the southwest, the reanalyses all show regional moistening centered in the southeast (Fig. 1, Fig. 5a–b). Based on CAMS, this trend in the reanalysis is associated with increases in ozone below the tropopause and CO above the tropopause (Fig. 5d,f). Increases in water vapor (Fig. 5b) and CO

below the tropopause are centered in the southeastern part of the domain, possibly suggesting an enhanced convective source to



**Figure 5.** Top faces: horizontal structure of deseasonalized anomalies in (a,b) PCWV, (c,d) PCO3, and (e,f) PCCO regressed onto the linear trend component of the first mode ($PC1_{TREND}$) based on Aura MLS. Side faces: vertical structure of deseasonalized anomalies in (a,b) water vapor, (c,d) ozone, (e,f) carbon monoxide averaged meridionally over 20°N–25°N (south face) and zonally over 30°E–130°E (east face) regressed onto $PC1_{TREND}$. Variables in the left column are from Aura MLS and variables in the right column are from CAMS; all are regressed onto $PC1_{TREND}$ from Aura MLS. The location of the Tibetan Plateau is marked by a dark grey contour and the boundaries of the east–west transect are marked by white lines in all panels. Stippling on the top face indicates locations where regression slopes are significant at the 95% confidence level.





the tropopause layer in this region. However, reanalysis-based trends in water vapor do not match those in Aura MLS (Fig. 5a). Instead, the satellite observations suggest a negative trend in water vapor below the tropopause, especially in the southwestern quadrant of the anticyclone. Aura MLS further indicates that these decreases in the lower part of the tropopause layer are accompanied by decreases in CO and increases in ozone below the tropopause, while increases in lower stratospheric water
vapor are linked to decreases in ozone and increases in CO above the tropopause (Fig. 5a,c,e). Trends based on CAMS indicate a very different distribution of changes in CO (Fig. 5f). Ozone trends are in somewhat better agreement (Fig. 5d), but this is not a useful litmus test because CAMS, like most of the reanalyses, assimilates Aura MLS ozone (Inness et al., 2019; Wright et al., 2025).

The S-RIP Final Report (SPARC, 2022) advises that trends based on reanalyses should be assessed with respect to three
criteria: (1) consistency across reanalyses; (2) consistency with (ideally independent) observations; and (3) a clear and convincing physical explanation. The trend discussed here meets the first criterion but does not satisfy the second. On one hand, this disagreement between the reanalyses and Aura MLS is qualified by the v5 water vapor retrieval for Aura MLS having been designed to remove a spurious increasing trend in water vapor (Livesey et al., 2021, 2022), which raises the possibility that these adjustments could have also removed physically meaningful regional trends over the MLS record. On the other hand,
except for ozone (for which CAMS assimilates retrievals from Aura MLS), trends in other variables based on Aura MLS also differ substantially from the patterns shown for CAMS (Fig. 5).

Figure 6 addresses the third criterion, as it is helpful to know the mechanisms behind the reanalysis-based trend even if this trend is spurious. Regressions shown in Fig. 6 are all calculated against $PC1_{TREND}$ from Aura MLS, thus removing the effects of different magnitudes of the year-to-year increase in PC1 across the reanalyses. Based on ERA5, warm season cloud ice water
content and high cloud cover increased over southern China, northern India, and the southeastern Tibetan Plateau during 2005–2021 (Fig. 6a). Detrainment from this deeper convection, also supported by increases in CAMS CO in this sector (Fig. 5f), may explain local increases in PCWV (Fig. $1b_E$–$b_S$), especially those centered in the lower part of the tropopause layer (Fig. 5b). Temperatures decreased in the lower stratosphere in this quadrant, although the magnitude of these decreases differs across the reanalyses (Fig. 6a, c–e). Increases in geopotential height and Montgomery streamfunction (MSF, Fig. 6b) are consistent
with expectations under global warming (see also the regional expression of greenhouse gas-driven tropospheric warming and stratospheric cooling in Fig. 6a). The local enhancements of MSF and geopotential height in the southeastern quadrant of the monsoon anticyclone imply additional tropospheric warming there, suggesting that the reduction in CPT temperature may result from an increase in CPT height. A colder CPT should lead to decreases rather than increases in water vapor above the cold point (Fig. 5a). However, the largest increasing trends in the reanalyses are located in the southeastern quadrant where
cooling of the CPT is largest (column (b) of Fig. 1).

Despite the qualitative consistency in the spatial pattern of the water vapor trend among the reanalyses, there are some notable differences in the spatial patterns. For example, whereas JRA-3Q (Fig. $1b_J$) shows moistening mainly over southern China, CAMS (Fig. $1b_C$) and ERA5 (Fig. $1b_E$) show significant increases westward over northern India. This difference may be understood to leading order from differences in the trend of CPT temperatures between ERA5 and JRA-3Q (Fig. 6c–e).
M2-SCREAM (Fig. $1b_S$) and MERRA-2 (Fig. $1b_M$) show even more widespread moistening spanning westward from the

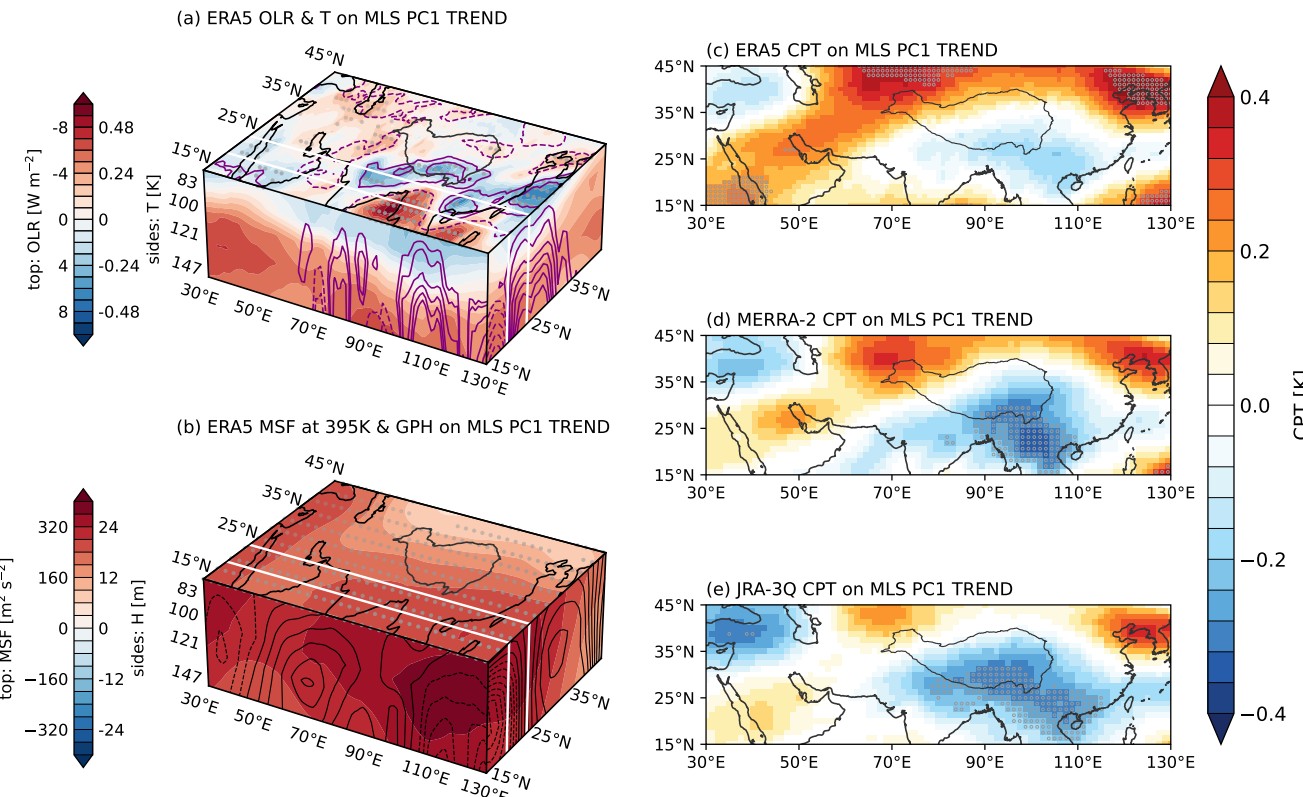

**Figure 6.** Left column: horizontal structure of changes in (a) outgoing longwave radiation (OLR; top face) and temperature (side faces) averaged meridionally over 20°N–25°N (south face) and zonally over 30°E–130°E (east face) over 2005–2021; and (b) as in (a), but for Montgomery streamfunction (MSF) on the $\theta = 395\,\mathrm{K}$ isentropic surface (top face) and geopotential height (side faces). Purple contours in panel (a) indicate anomalies in high cloud fraction (top face; intervals of 2 percentage points from ±2%) and cloud ice water content (side faces; intervals of $0.2\,\mathrm{mg\,kg^{-1}}$ from $\pm0.1\,\mathrm{mg\,kg^{-1}}$). Black contours in panel (b) indicate changes in meridional (south face) and zonal (east face) winds at intervals of $0.2\,\mathrm{m\,s^{-1}}$. All variables are regressed onto the linear trend component of the first mode (PC1$_{\mathrm{TREND}}$) based on Aura MLS. Right column: changes in cold point tropopause (CPT) temperatures based on (c) ERA5, (d) MERRA-2, and (e) JRA-3Q regressed onto PC1$_{\mathrm{TREND}}$. The location of the Tibetan Plateau is marked by a dark grey contour in all panels and the boundaries of the east–west transect are marked by white lines in panels (a) and (b). Stippling indicates locations where regression slopes are significant at the 95% confidence level based on Student's $t$ test.





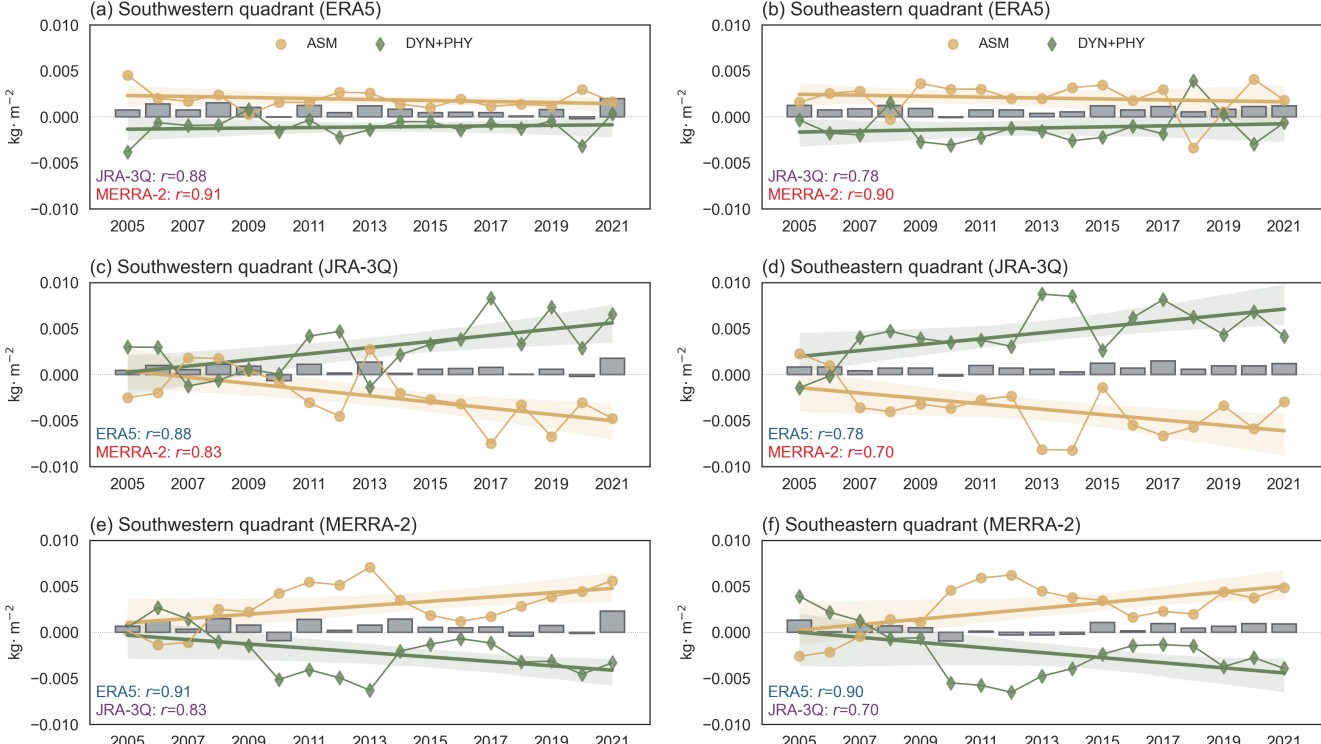

**Figure 7.** Yearly variations in the sum of the dynamics and physics terms (green lines), assimilation increments (yellow lines), and time rate of changes in partial column water vapor (gray boxes) in the monsoon tropopause layer based on (a,b) ERA5, (c,d) JRA-3Q, and (e,f) MERRA-2 over (a,c,e) the southwestern quadrant (15°N–25°N,30°E–90°E) and (b,d,f) the southeastern quadrant (15°N–25°N,90°E–130°E) of the monsoon anticyclone. Correlation coefficients between net water vapor tendency time series based on individual reanalyses are listed in the lower left corner of each panel.

South China Sea (larger magnitudes in the MERRA-2 signal are largely attributable to the weaker slope of PC1$_{\text{TREND}}$ in this reanalysis; Fig. 3b). These differences are more difficult to attribute to physical processes because M2-SCREAM assimilates Aura MLS v4 water vapor retrievals and MERRA-2 damps variability in lower stratospheric water vapor by relaxing to a mean seasonal cycle. However, it is worth noting that MERRA-2 produces a pattern of cold point temperature trends that lies

roughly between that in ERA5 and that in JRA-3Q, with a similar spatial pattern (Fig. 6c–e). The robust nature of these CPT temperature changes, which are influenced by the model physics and vertical resolution but also relatively well constrained by observational data assimilation (Tegtmeier et al., 2020), are consistent with increased convection and high cloud cover over the southeastern Tibetan Plateau and the areas to its southeast.

Trends in reanalysis variables may also result from jumps or drifts due to changes in assimilated observations (SPARC,

2022). The budget decomposition articulated in Eq. (2) provides an opportunity to evaluate this possibility. Figure 7 shows year-to-year variations in specific humidities due to resolved transport (DYN; $-\nabla \cdot (\mathbf{V}q) - \frac{\partial(\omega q)}{\partial p}$), parameterized physics




(PHY; $S_{\mathrm{phy}}$), and data assimilation (ASM; $S_{\mathrm{ana}}$) over 2005–2021 based on ERA5, JRA-3Q, and MERRA-2. We split the monsoon anticyclone domain into quadrants covering the southeastern (90–130°E, 15–25°N), southwestern (30–90°E, 15–25°N), northeastern (90–130°E, 25–45°N), and northwestern (30–90°E, 25–45°N) parts of our analysis domain. We focus
especially on the southeastern quadrant, since this part of the anticyclone shows the largest increasing trends in UTLS water vapor over 2005–2021; time series for the northwestern and northeastern quadrants are provided in the online supplement (Fig. S4). Across the three reanalysis datasets, trends in dynamical and physical tendencies are generally compensated by data assimilation. However, the trends in these individual components are not consistent across the products. Both dynamical and physical tendencies in the southwest and southeast quadrants based on ERA5 were relatively flat during 2005–2021,
with negative values on average. By contrast, JRA-3Q shows an increasing trend with predominantly positive values, while MERRA-2 shows a decreasing trend with predominantly negative values (Fig. 7). Regional moistening over the warm season shows no evidence of a trend in any of the reanalyses, indicating that the sources of the reanalysis-based increasing trends are outside the monsoon region or season and thus beyond the scope of this study. Long-term warming of the tropical cold point tropopause, as recently reported by Zolghadrshojaee et al. (2024), provides a possible explanation. Such a trend could be
consistent with the trends based on both Aura MLS and the reanalyses in the lower stratosphere (Fig. 5a–b). As such, trends in this part of the tropopause layer do satisfy the three criteria: (1) consistency across reanalyses, (2) consistency with independent observations, and (3) a plausible physical mechanism. Overall, although we find several reasons to doubt the reanalysis-based trends in water vapor below the tropopause, a more complete analysis of trends in UTLS composition and related fields in this region is warranted.

## 3.2  The interannual mode

Figure 8a further illustrates how we divide interannual variability in PC1 based on Aura MLS into trend (PC1$_{\mathrm{TREND}}$; Fig. 3b) and interannual (PC1$_{\mathrm{IAV}}$; Fig. 3c) components. For PC1$_{\mathrm{IAV}}$, we define moist years as those for which the 25$^{\mathrm{th}}$ percentile of pentad-mean PC1$_{\mathrm{IAV}}$ was positive and dry years as those for which the 75$^{\mathrm{th}}$ percentile was negative. Applying this definition to PC1$_{\mathrm{IAV}}$ from Aura MLS over 2005–2021, six years were anomalously moist, six years were anomously dry, and five years
were neutral (all within the last six years of the record; Fig. 8b). Remarkably, PC1$_{\mathrm{IAV}}$ maintained the same sign through all 31 pentads in seven of the seventeen years (dry: 2006, 2012, 2013; moist: 2007, 2010, 2011, 2017). These results reinforce the impression from Fig. 3 that the sign of PC1$_{\mathrm{IAV}}$ is set before the monsoon and maintained throughout the monsoon season.

Candidate mechanisms for controlling water vapor in the monsoon tropopause layer at interannual time scales include the El Niño–Southern Oscillation (ENSO) and the quasi-biennial oscillation (QBO). To represent ENSO variability, we use
the Oceanic Niño Index, defined as 3-month rolling mean sea surface temperature anomalies in the Niño3.4 region (190°E–240°E, 5°S–5°N). To represent QBO variability, we use the updated Berlin QBO time series at 50 hPa and 70 hPa, which is based on radiosonde observations from Singapore (1.34°N, 103.89°E) during this period. Figure 8c shows lag correlations of these indices from the preceding August to the concurrent July against seasonal-mean PC1$_{\mathrm{IAV}}$ (grey diamonds in Fig. 8b). Correlations are significant for both QBO indices, with the largest correlations ($r > 0.7$) found for the December (50 hPa)
and March (70 hPa) preceding monsoon onset. These positive correlations are consistent with the moistening influence of





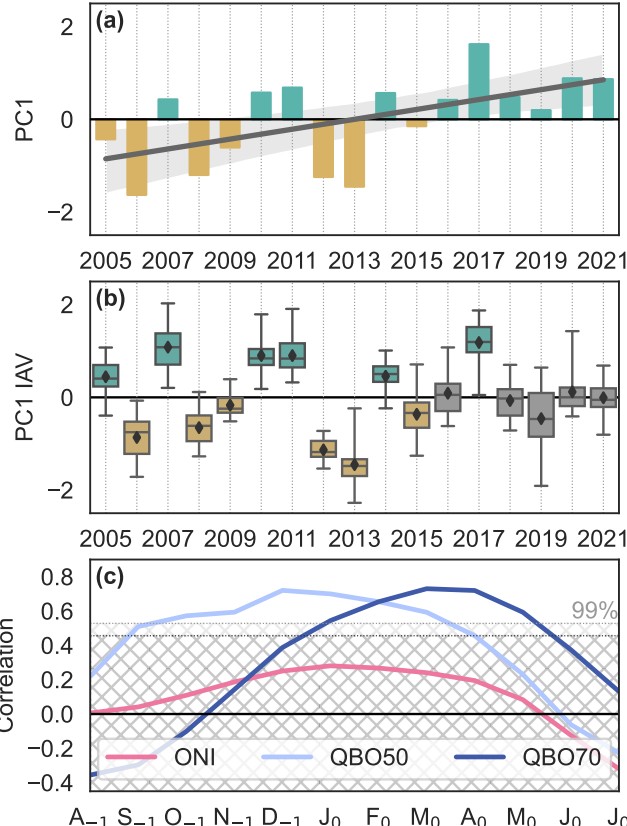

**Figure 8.** Yearly variations in (a) the first principal component (PC1) of water vapor variability in the monsoon tropopause layer based on Aura MLS (Fig. 3a) and (b) the interannual variability component of PC1 (Fig. 3c). (c) Lag correlations of interannual mean $PC1_{IAV}$ (grey diamonds in panel b) against the Oceanic Niño Index (ONI; pink) and quasi-biennial oscillation (QBO) indices at 50 hPa (light blue) and 70 hPa (dark blue) from the previous August to the concurrent July.

warmer tropical tropopause temperatures as the QBO westerly phase descends toward the tropopause (Geller et al., 2002), particularly during Northern Hemisphere winter and spring (Randel and Jensen, 2013). Correlations with ENSO are weakly positive and are not statistically significant over this period. Correlations between $PC1_{IAV}$ and commonly used indices of the tropospheric monsoon intensity, including the All-India Rainfall Index produced by the India Meteorological Department and

the tropospheric zonal wind shear index proposed by Webster and Yang (1992), are essentially zero ($|r| \leq 0.05$).

Figure 9 shows the vertical and horizontal structure of deseasonalized anomalies in water vapor and CO based on CAMS, along with temperature, cloud fields, MSF, geopotential height, and OLR based on ERA5. All variables are regressed onto Aura MLS $PC1_{IAV}$. Anomalies in water vapor and CO are reported as fractional changes given the sharp vertical gradients in these species around the tropopause. The horizontal structure of fractional water vapor anomalies based on CAMS (Fig. 9c)

shows a regional-scale positive anomaly of about 5%, with the largest anomalies arcing in a band from the Persian Gulf along





the northern edge of the Tibetan Plateau to the East Asian monsoon region. This distribution is consistent with the absolute anomalies in PCWV shown in column (c) of Figure 1. The southwestern segment of this band, spanning from the Persian Gulf northeastward over the western Tibetan Plateau, also shows positive anomalies in high cloud fraction and negative anomalies in OLR (Fig. 9a). These indicators of enhanced convective activity correspond to negative anomalies in radiative heating on

the 390 K potential temperature surface (Fig. S5a) and positive anomalies in non-radiative (mainly convective) heating in the lower part of the tropopause layer (Fig. 4a–b; Fig. S5c). Cloud ice, cloud cover, and diabatic heating anomalies thus show that the moist phase of $PC1_{IAV}$ is associated with stronger convective activity over the Indus River valley in a band that extends northeastward from the Arabian Sea to the Karakoram Gap. The vertical structure along the southern flank of the anticyclone likewise shows large positive anomalies of convective heating penetrating into the tropopause layer in the western part of the

domain and near-zero anomalies in the eastern part of the domain (Fig. 4a). In the zonal mean, the largest positive anomalies are located north of 25°N (Fig. 4b), consistent with the second center of enhanced water vapor located over the East Asian monsoon region (Fig. 1c; Fig. 9c). Positive water vapor anomalies are most widespread at 68 hPa; however, unlike anomalies below the tropopause, positive anomalies at 68 hPa are strongest in the southeastern quadrant. This difference is consistent with either in-mixing of moist air from the anomalously warm tropical lower stratosphere or clockwise ascent of water vapor

anomalies over East Asia in the 'upward spiraling' carousel of the anticyclone (e.g. Vogel et al., 2019; Konopka et al., 2023), and highlights the persistence of anomalies associated with this mode. Results based on other reanalyses are generally similar to those based on CAMS (Fig. S6). MERRA-2 is again the outlier, as its strong relaxation of stratospheric water vapor to a climatology eliminates almost all interannual variability at 68 hPa. Greater consistency in the magnitudes of relative anomalies compared to those of absolute anomalies indicates that reanalysis products with larger interannual variability also tend to have

larger mean humidities at these levels.

Positive anomalies in MSF and geopotential height (Fig. 9b) indicate a warmer upper troposphere along the southern flank of the anticyclone, consistent with arrival of the QBO westerly phase at the tropopause. Weak warm anomalies at lower levels to the south of 25°N and weak cool anomalies to the north (Fig. 9a) imply westerly anomalies through most of the region (Fig. 9b) by thermal wind balance. Significant ozone anomalies are only found along the southern flank of the anticyclone in

the lower stratosphere (Fig. S7). The distribution of ozone anomalies is consistent among five of the six datasets, with JRA-3Q the exception (Fig. S7). JRA-3Q, which does not assimilate vertical ozone profiles (Kosaka et al., 2024; Wright et al., 2025), shows significant increases in ozone within the anticyclone associated with this mode, with large signals centered just below the tropopause (Fig. S7b). CO concentrations are significantly reduced over northern India and the Bay of Bengal and increased southeastward of the East Asian monsoon region. These anomalies are consistent among the CO products based on Aura MLS,

MERRA-2, and CAMS (Fig. S8a–c).

Water vapor anomalies associated with $PC1_{IAV}$ arc clockwise around the anticyclone (right column of Fig. 1), with the minimum located over the Bay of Bengal and the maximum located over East Asia and the western Pacific. Figure 9 provides a framework for understanding this pattern. First, systematically deeper convection in the southwestern quadrant of the monsoon domain (Fig. 4a and Fig. 9a) injects larger amounts of water above the LZRH (Wright et al., 2025). This water-rich air is

advected northward around the inner western side of the anticyclone, where strong lower stratospheric radiative heating rates

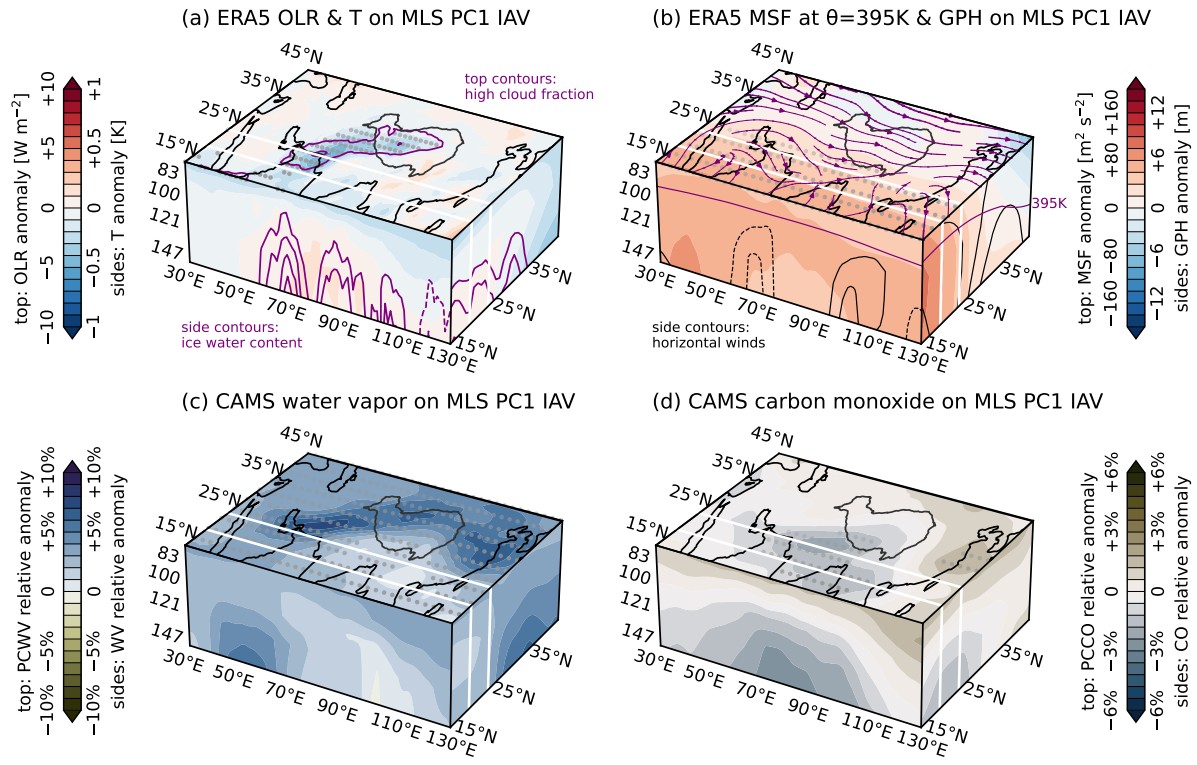

**Figure 9.** (a) Horizontal structure of deseasonalized anomalies in outgoing longwave radiation (OLR; top face) and vertical structure of deseasonalized anomalies in temperature (side faces) along with high cloud fraction and cloud ice water content (contours; averaging domains and contour intervals as in Fig. 6a); (b) as in (a) but for Montgomery streamfunction (MSF) and winds (streamlines) on $\theta = 395\,\mathrm{K}$ (top face) and geopotential height (shading) and horizontal winds (contours; intervals as in Fig. 6b); (c) as in (a) but for partial column water vapor (PCWV; top face) and specific humidity (side faces); and (d) as in (a) but for partial column CO (PCCO) and CO mass mixing ratios (side faces). All variables are regressed onto the interannual component of the first mode (PC1$_{\mathrm{IAV}}$) based on Aura MLS (Fig. 3c). Variables in the top row are from ERA5 and variables in the bottom row are from CAMS. The location of the Tibetan Plateau is marked by a dark grey contour and the boundaries of the east–west transect are marked by white lines in all panels. The vertical location of the 395 K isentropic surface is indicated by the purple line on the side faces of panel (b). Stippling indicates locations where regression slopes are significant at the 95% confidence level based on Student's $t$ test.



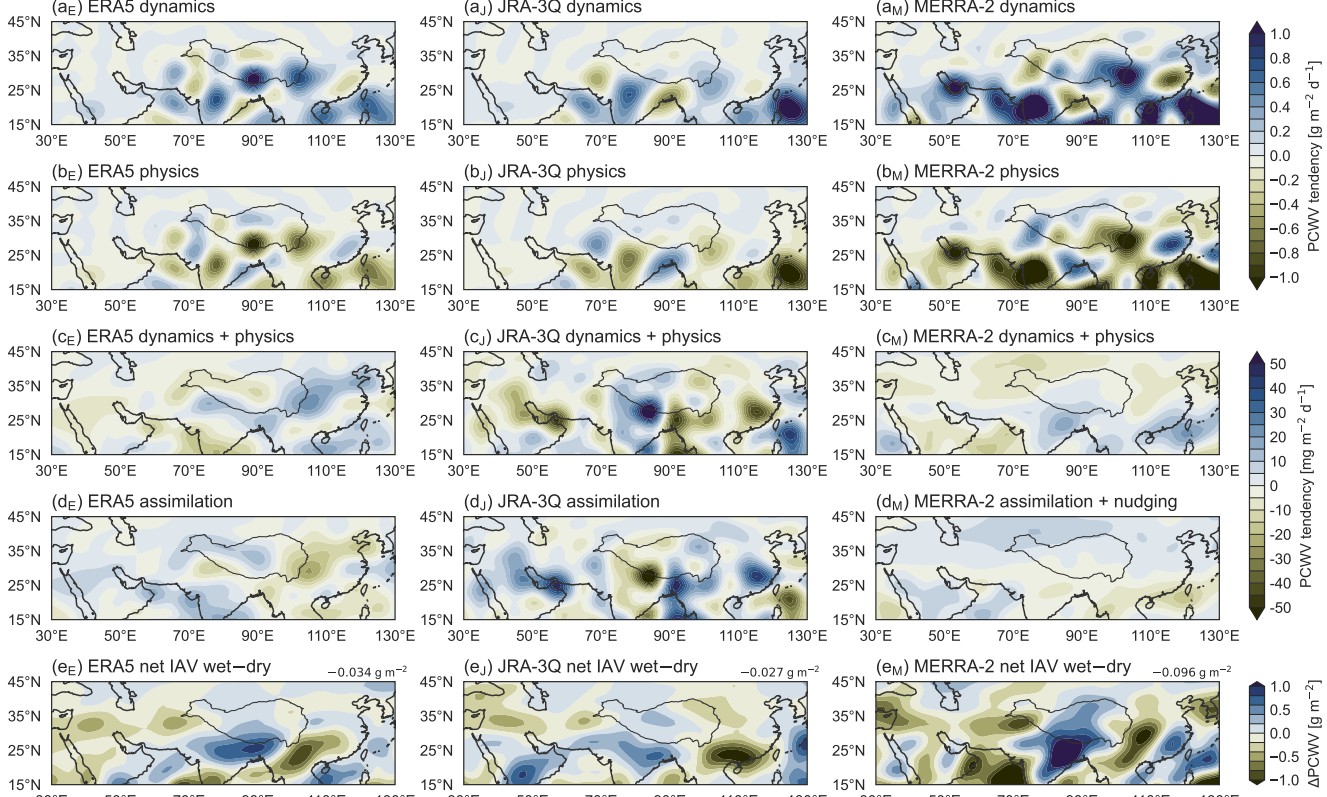

**Figure 10.** Differences of composite mean water vapor tendencies due to (a) dynamics (i.e. moisture flux convergence), (b) parameterized physics, (c) dynamics and parameterized physics, (d) data assimilation, and (e) net moisture sources for ($a_E$–$e_E$) ERA5, ($a_J$–$e_J$) JRA-3Q, and ($a_M$–$e_M$) MERRA-2 between moist years and dry years based on the first principal component detrended interannual variability (PC$_{IAV}$, Fig. 8b). The location of the Tibetan Plateau is marked by a dark grey contour in all panels. Net changes in area-mean tropopause-layer PCWV are listed in the upper right corner of each panel.

relative to the monsoon core balance ascent across isentropic surfaces (see Tegtmeier et al., 2022, their Figs. 8.59–8.60). The moist phase of PC1 can thus be explained by anomalously deep convection delivering large amounts of water to the tropopause layer in the southwestern part of the domain and subsequent upward spiraling transport. Conversely, the dry phase is linked to negative initial anomalies due to a colder tropical tropopause in the pre-monsoon, weaker convection in the southwest and stronger convection in the southeast, and colder temperatures along the southern flank of the anticyclone that effectively throttle 420 moist air detrained in the southeast. Additional factors may include variations in cross-tropopause transport over the East Asian summer monsoon (e.g. Luo et al., 2013) or eastward eddy shedding (e.g. Popovic and Plumb, 2001; Honomichl and Pan, 2020), both of which could contribute to the local maximum in water vapor anomalies in the northeastern part of the domain (Fig. 9c).

Figure 10 illustrates contributions to PC1$_{IAV}$ based on Eq. (2) applied to ERA5, JRA-3Q, and MERRA-2. These distri-425 butions are based on the composite mean over the six moist years minus the composite mean over the six dry years. In all





three reanalyses, the dominant balance associated with this mode is between moistening by moisture flux convergence and drying due to parameterized physical processes (Fig. 10a$_E$–b$_M$; see also Wright et al., 2025). The dynamical moistening and physical drying effects nearly offset each other, with the sum of these two terms smallest in MERRA-2 and largest in JRA-3Q (Fig. 10c$_E$–c$_M$). The three reanalyses show different distributions of net moisture sources to the tropopause layer, with
ERA5 highlighting the Sichuan Basin, JRA-3Q highlighting northern India and the adjacent south slope of the Himalayas, and MERRA-2 highlighting the coastal regions around the Bay of Bengal (Fig. 10e$_E$–e$_M$). However, all three reanalyses are consistent in producing sources of water vapor to the tropopause layer over the South China Sea. As with the trend, the sum of the dynamics and physics terms is again largely compensated by the effects of data assimilation in each system (Fig. 10d$_E$–d$_M$). After accounting for data assimilation effects, the net change in area-mean tropopause-layer PCWV is negative over the
warm season in all three reanalyses, consistent with relaxation of the initial pre-monsoon anomaly. Several anomalies in the net tendencies are robust across the reanalyses. For example, robust positive tendencies during moist years are located over the western North Pacific and South China Sea, as well as the Indo-Gangetic Plain between the Bay of Bengal and the Indus River Valley, while robust positive tendencies during dry years are concentrated mainly over the Yunnan Plateau, between the South China Sea and the Bay of Bengal, and the western coast of southern India.

**3.3   The subseasonal modes**

Figure 11 shows the implied quadratic cycle of PC2 and PC3, which together describe at least 10–20% of water vapor subseasonal variability in the Asian monsoon tropopause layer. Anomalies in OLR, geopotential height on the 100 hPa isobaric surface, and wave activity fluxes following Takaya and Nakamura (2001) are shown for the transition periods between different phases. Not all variability in PC2 and PC3 evolves through this coupled cycle, as discussed below: many different modes of
convective variability share similar centers of action in this region (e.g. Goswami and Mohan, 2001). However, the cycle serves as a useful framework for investigating the mechanisms behind these two modes. Fractions of events that propagated from the previous step in the cycle (Fig. 11a-i to d-i) or to the next step in the cycle (Fig. 11a-ii to d-ii) within two pentads (10 days) during 2005–2021 are marked in the upper right corners of the transition maps. The strongest links are between the moist phase of PC3 (PC3+), the moist phase of PC2 (PC2+), and the dry phase of PC3 (PC3−), clockwise from the top to the bottom of
Fig. 11 (see also Fig. S9 in the online supplement).

     Water vapor anomalies associated with PC3+ exhibit a horizontal dipole structure with a slight southwest-to-northeast tilt (Fig. 11a; see also Fig. 2b). Unlike the interannual mode, which is centered above the tropopause in the lowermost stratosphere, anomalies associated with this mode are mainly located in the lowest part of the tropopause layer. Positive anomalies centered in the southwest are linked to dry anomalies over the northeastern Tibetan Plateau, with amplitudes that decrease sharply with increasing height. Conversely, water vapor anomalies associated with PC3−, discussed in detail below, are associated
with moist anomalies over the northern Tibetan Plateau and weak dry anomalies over the Arabian Peninsula and Persian Gulf (Fig. 11c). The largest moist anomalies associated with PC2+ are centered over Iran, Pakistan, and adjacent seas, similar to those in PC3+. However, in PC2+, these anomalies also connect through the ridgeline of the anticyclone to anomalies of the same sign northeast of the Tibetan Plateau above the East Asian monsoon rainband (Fig. 11b). Moist anomalies associated with



**Figure 11.** (a)–(d) Cycle of water vapor sub-seasonal variability based on CAMS showing a typical sequence of transitions among PC2 and PC3, along with (a-i)–(d-ii) deseasonalized anomalies in outgoing longwave radiation (OLR, shading) based on CERES and 100-hPa geopotential height (Z100; contours at $\pm 15$ m) and wave activity fluxes (WAF100; vectors for magnitudes $> 2\,\mathrm{m}^2\,\mathrm{s}^{-2}$) based on ERA5 during transitions. For example, panel (a-i) shows distributions of anomalous OLR, anomalous Z100, and mean WAF100 for all PC3+ events that follow PC2− events within two pentads and panel (a-ii) shows distributions of the same variables for all PC3+ events that lead PC2+ events within two pentads. Z100 and water vapor anomalies in panels (a)–(d) are averaged over all events. Z100 contours (a-i) to (d-ii) are dark red for the following phase and light red for the leading phase, with WAF100 vectors shown for locations where mean wave activity flux divergence during the transition exceeds $\pm 1 \times 10^{-5}\,\mathrm{m}\,\mathrm{s}^{-2}$. OLR anomalies are averaged over pentads –2 and –1. (e) Co-evolution of PC2 and PC3 from 15 July (marked by a triangle) to 13 September 2013 (square). The start of each pentad is marked by a circle.





PC2− (Fig. 11d) are centered over the southeastern Tibetan Plateau, bracketed by dry anomalies to the northwest and northeast and with amplitudes considerably weaker than those associated with the other phases. The moist anomaly over the southeastern Tibetan Plateau during PC2− produces an anomaly of the opposing sign when PCWV is regressed onto PC2 (Fig. 2a), even though this anomaly is not evident in the PC2+ composite (Fig. 11b).

It is important to emphasize that the distributions shown in Fig. 11 are based on composites for standardized principal

components exceeding ±1, whereas those shown in Fig. 2 are regressed onto the principal components (see Figs. S10 and S11 for three-dimensional distributions regressed onto PC2 and PC3, respectively). The tropopause layer above the monsoon is very dry in absolute terms. Positive anomalies thus tend to be larger in magnitude than negative anomalies, as illustrated by Fig. 11. Accordingly, the two poles of the east–west dipole shown in Fig. 2b manifest more as individual monopoles in Fig. 11a and Fig. 11c, with the opposing dry anomalies almost disappearing. Water vapor anomalies associated with PC2 and PC3 are

robust across the reanalyses and in good agreement with Aura MLS (see Figs. S10 and S11).

As noted above, the North test indicates that PC2 and PC3 are mutually independent. However, as illustrated by Fig. 11, lead–lag correlations suggest that these modes often occur in quadrature. PC3+ precedes PC2+ by one or two pentads 40–50% of the time, with a strong area-weighted pattern correlation ($r = 0.91$) between PCWV anomalies composited on PC2+ and average PCWV anomalies two pentads after PC3+ (Fig. S9a–c). PC3− then follows PC2+ within two pentads more than

50% of the time. Accordingly, the pattern of PCWV anomalies composited on PC2+ is also highly correlated with average PCWV anomalies 2.5 pentads before PC3− ($r = 0.87$; Fig. S9c–e). PC3− is often followed by strong wave breaking along the eastern flank of the anticyclone and enhanced precipitation over southern China (Fig. 11c-ii). These processes may complicate the transition from PC3− to PC2−, which only occurs about 30% of the time within 10 days and shows weaker lead–lag pattern correlations ($r = 0.69$; Fig. S9e–g). Evidence that PC2− is followed by PC3+ about 30–40% of the time completes the

quadratic cycle of water vapor subseasonal variability (Fig. 11d), although pattern correlations between PC2− and anomalies preceding PC3+ are again relatively weak ($r = 0.55$; Fig. S9g,h,a). Figure 11e shows an example trajectory in the PC2–PC3 phase space, in this case calculated for 15 July–13 September 2013. The trajectory is based on bandpass-filtered 3-hourly CAMS water vapor anomalies projected onto PC2 and PC3 from Aura MLS. The primarily clockwise evolution of the trajectory matches the 'typical' sequence illustrated in Fig. 11a–d.

The timescale of the PC3+ to PC3− part of the cycle (15–25 days) slightly exceeds that corresponding to quasi-biweekly variations in monsoon circulation and precipitation fields (Ortega et al., 2017; Ren et al., 2019), although this may arise in part from our use of pentad-mean anomalies. Indeed, the return periods indicated by Fig. 11e are somewhat shorter at 10–15 days. Monsoon quasi-biweekly waves have been shown to exert substantial influences on UTLS composition in this region (e.g. Randel and Park, 2006; Satheesh Chandran and Sunilkumar, 2024). Nonetheless, the relatively tenuous links to PC2− and

the substantial percentages of PC2+ events that are neither preceded by PC3+ nor followed by PC3− emphasize that these modes also vary independently. Convective activity fluctuates on many time scales within the Asian monsoon, from diurnal pulses (e.g. over the southeastern Tibetan Plateau and other high altitude areas; Hirose and Nakamura, 2005; Zhao et al., 2022), to the synoptic scale (e.g. monsoon depressions; Hunt et al., 2016; Dong et al., 2017) upward to the quasi-biweekly and 30–50 day intraseasonal scales (e.g. the boreal summer intraseasonal oscillation; Chatterjee and Goswami, 2004; Annamalai




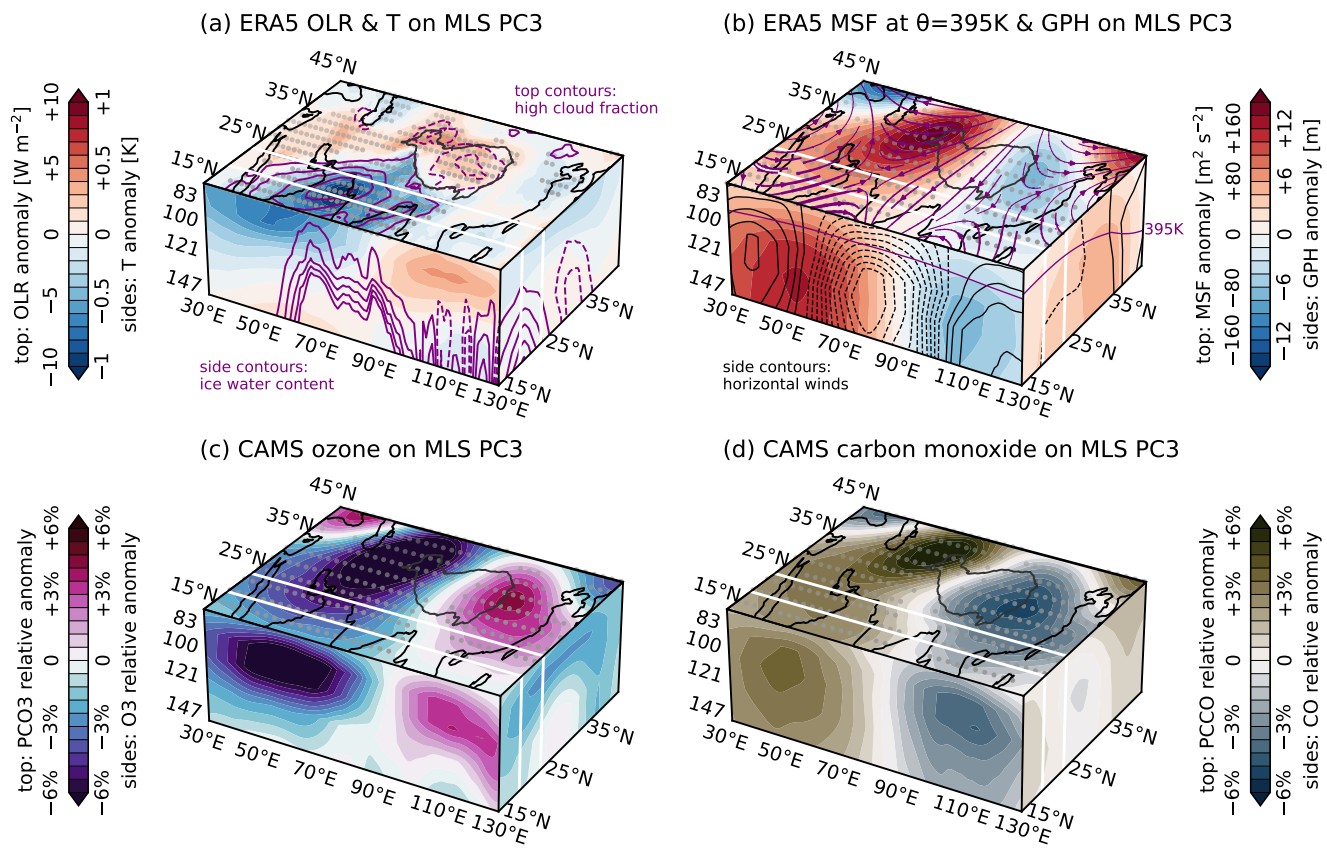

**Figure 12.** As in Fig. 9, but for PC3 from Aura MLS (Fig. 3e).

and Sperber, 2005; Ortega et al., 2017). Variability of water vapor near the tropopause that projects onto PC2 and PC3 may emerge when these different modes of organized convective variability move into or through centers of action similar to those highlighted by the OLR anomalies in Fig. 11. Phase space trajectories for periods dominated by other convective modes evolve differently from those shown in Fig. 11e, and thus may provide a means of distinguishing the imprints of these myriad modes of monsoon convective variability on water vapor near the tropopause.

Anomalies in OLR, geopotential height at the 100 hPa level, and wave activity fluxes provide insight into the mechanisms involved in this cycle. As the links are strongest between the two phases of PC3 and PC2+, we start from PC3+, at the top of the cycle in Fig. 11. To support the discussion, we also link these transitional anomalies to anomalous temperatures, upper-level cloud fields, OLR, geopotential height, winds, MSF, ozone, and CO regressed onto PC3 (Fig. 12) and PC2 (Fig. 13). An anomalous high west of the Tibetan Plateau emerges during the days leading up to PC3+ (Fig. 11a-i), with possible triggers including anomalous convection over northern India and tropopause folding upstream of the Plateau. This intensified



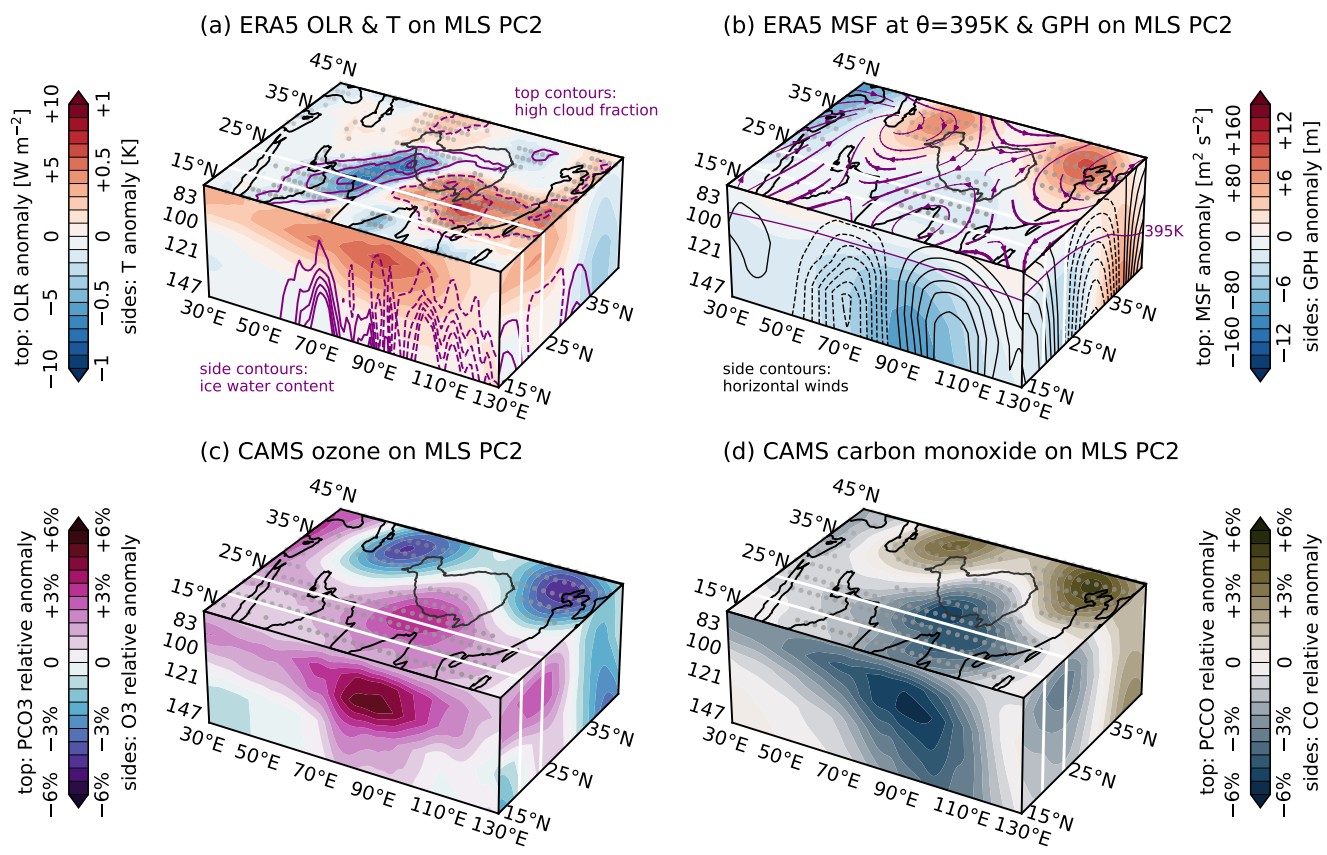

**Figure 13.** As in Fig. 9, but for PC2 from Aura MLS (Fig. 3d).

anticyclonic center in the west persists through PC3+, by which point its northern edge overlaps with the subtropical jet (Figs. 11a and 12b). Oscillations of the jet that result from dynamical restoring of this poleward displacement upstream of the Tibetan Plateau produce a companion anomaly downstream over northern China. Each center spans approximately 20–25° longitude, separated by most of the width of the Tibetan Plateau (also ∼25° longitude). These distances indicate a zonal

wavenumber of 8–9. Given a background flow of 25–30 m s−1, the anomaly over northern China would be expected to emerge within a pentad of the forced anomaly west of the Tibetan Plateau, in line with the timescales shown in Fig. 11. Convection over the Indus River valley remains intense into PC2+ (Fig. 13a), with a tail of enhanced high cloud cover and reduced OLR suggestive of the 'up-and-over' transport mechanism that often links lowland monsoon depressions to episodes of strong summer precipitation over the southwestern Tibetan Plateau (Dong et al., 2016, 2017)

During PC2+, moist anomalies in the southwest (30°E–70°E along 25°N) are associated with positive anomalies in cloud ice water content (Fig. 13a) and total diabatic heating (Fig. 4c). By contrast, cloud ice water content and total diabatic heating are





reduced in the southeast (90°E–130°E) during PC2+. These anomalies are qualitatively robust among the reanalyses (Figs. S2, S10, and S12). The reduced diabatic heating in the southeast is situated under an upper-level cyclonic anomaly over the Bay of Bengal, as shown by negative anomalies in MSF on the 395 K isentropic surface (Fig. 13b). Temperature anomalies in this

region are positive (Fig. 13a), indicating that the reduction in MSF results from a decrease in geopotential height (Fig. 13b). The decrease in geopotential height implies a cooler and shallower tropospheric column, consistent with reduced convective heating over the Bangladesh and southern China. MSF anomalies associated with the pair of vortices in the north are also linked to anomalous geopotential heights (Fig. 13b), as both regions feature cold temperature anomalies near the tropopause (Fig. 13a).

Warm temperature anomalies span the entire southern flank of the anticyclone near the tropopause during PC2+ (Figs. 13a and 14a), with the largest warming centered between positive cloud ice anomalies in the southwest and negative cloud ice anomalies in the southeast (Fig. 13a). This location corresponds to the climatological location of the monsoon cold trap (Wright et al., 2025, their Fig. 2). The variations in high clouds and attendant changes in OLR (Fig. 13a) further emphasize the shift in convective activity from the southeast to the southwest in this phase. Radiative heating anomalies on the 390 K isentropic

surface are opposite to those in temperature (Fig. S12a) and can therefore be interpreted via the 'Newtonian cooling' approximation (e.g. Wright and Fueglistaler, 2013), in which radiative heating rates are modeled as inversely proportional to the difference between ambient temperature and a fixed radiative equilibrium temperature.

Anomalies in ozone (Fig. 13c) and CO (Fig. 13d) reflect fluctuations in the upper-level anticyclone. Air with characteristics of the climatological anticyclone (low ozone and high CO) is concentrated in the pair of anticyclonic vortices to the northeast

and northwest of the Tibetan Plateau during PC2+, implying an overall northward shift in the position of the anticyclone (see also regional-mean zonal wind anomalies along the east face of Fig. 13b and changes in the 2 PVU potential vorticity contour in Fig. 14b). Air along southern flank exhibits more stratospheric characteristics (high ozone and low CO), especially around 90°E where the upper-level cyclonic anomaly is centered. These changes indicate that anomalous descent and compression contribute to the warming along the southern flank, in line with reduced lower stratospheric radiative heating (Fig. S12) and geopotential

height (Fig. 13b). Large potential vorticity anomalies extend along the southern flank of the anticyclone downstream of the reduced convection over the Bay of Bengal (Fig. 14b). This distribution implies a positive diabatic potential vorticity tendency (i.e. weaker horizontal divergence) over the Bay of Bengal, with the resulting positive anomalies advected westward by the tropical easterly jet.

Variations in water vapor associated with PC2 can be explained to leading order by changes in convective activity and temper-

atures near the tropopause (Randel et al., 2015), mediated by the response of the upper-level anticyclone to the redistribution of convective heating (Nützel et al., 2016; Siu and Bowman, 2019). Area-mean CPT temperatures are much warmer than normal during the PC2+. Positive anomalies of 0.3–0.5 K over much of the region south of 30°N (Fig. 14a and Fig. S14e–f) translate to an increase of around 5–9% in saturation mixing ratios at the CPT, sufficient to explain the positive fractional anomalies in water vapor anomalies west of this region (∼6–9% Fig. 11b). The largest water vapor anomalies are in the southwestern

quadrant, downstream of both the largest warm anomalies (Fig. 14a) and the enhanced deep convection north of the Arabian Sea (Fig. 13a, Fig. 11b-i). Stronger deep convection can inject water directly into the tropopause layer (Ueyama et al., 2018),



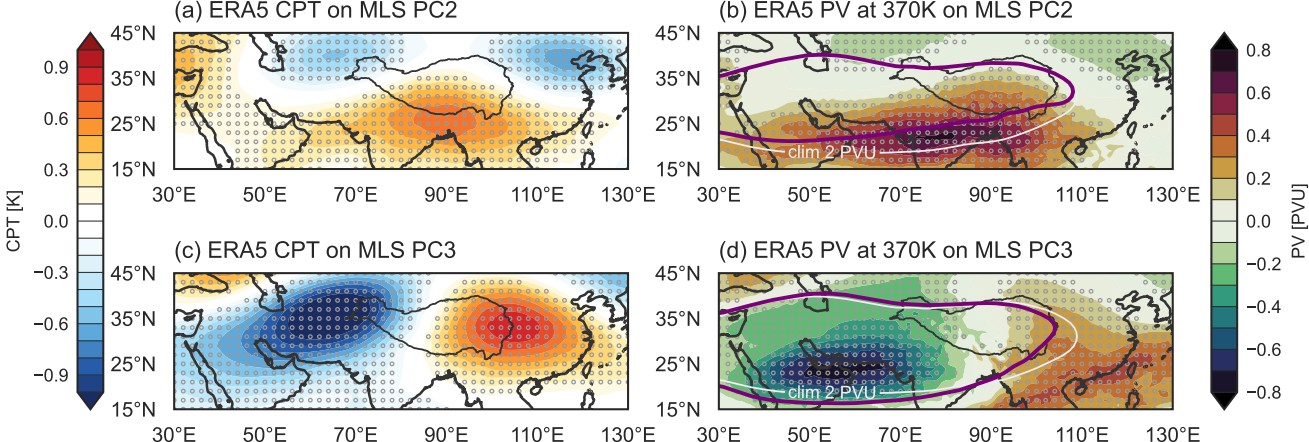

**Figure 14.** Deseasonalized anomalies of (a,c) cold point tropopause (CPT) temperatures and (b,d) potential vorticity (PV) on the 370 K isentropic surface based on ERA5 regressed onto (a)–(b) PC2 (Fig. 3d) and (c)–(d) PC3 (Fig. 3e) from Aura MLS. Dark grey contours in all panels mark the location of the Tibetan Plateau. White contours in (b,d) mark the 2 PVU contour ($1\,\mathrm{PVU} = 10^{-6}\,\mathrm{K\,m^2\,kg^{-1}\,s^{-1}}$) on the 370 K isentropic surface based on the average for July 2005–2020. Stippling indicates that regressions are significant at the 95% confidence level.

while warmer CPT temperatures allow more of this injected water to remain. However, the anomalous cyclone to the east and the anomalous anticyclone to the north restrict the flow of anomalously humid air around the climatological anticyclone. The moist center in the northeast coincides with the northeastern anticyclonic anomaly, where ozone is reduced and CO is

enhanced (Fig. 13c–d). These anomalies may thus be interpreted as an anticyclonic vortex displaced from the relatively humid southeastern quadrant into the drier northeast, possibly as a prelude to eastward eddy shedding (e.g. Honomichl and Pan, 2020; Siu and Bowman, 2020).

PC3− evokes the canonical 'Tibetan Plateau' mode of the upper-level anticyclone (Qian et al., 2002; Zhang et al., 2002; Nützel et al., 2016), which has previously been linked to coherent anomalies in UTLS composition above the ASM (e.g. Yan

et al., 2011; Kumar and Ratnam, 2021). The dry phase of PC3 follows a relatively quiescent period in terms of convective activity, with no notable negative anomalies in OLR within the analysis domain (Fig. 11b-ii,c-i). During this phase, the northeastern high from PC2+ propagates southward along the eastern flank of the anticyclone, as shown by enhanced southward wave activity fluxes over much of China east of the Tibetan Plateau (Fig. 11b-ii,c-i). Accordingly, the dry phase of this mode is associated with increased MSF in the east and decreased MSF in the west (Fig. 12b). Changes in MSF again arise mainly from

changes in geopotential height rather than local temperature anomalies (Fig. 12a–b), with reduced MSF anomalies associated with warmer temperatures and vice versa. Changes in ozone (Fig. 12c) and CO (Fig. 12d) likewise show that the active (anticyclonic) pole contains air with more tropospheric characteristics, while the inactive (cyclonic) pole contains air with more stratospheric characteristics.





Unlike PC2, variations in local dehydration cannot provide a direct explanation for variations in water vapor in this mode,
because CPT temperatures are anomalously cold where water vapor is enhanced and warm where water vapor is reduced (cf.
Fig. 14c, Fig. 11a). Concurrent variations in both isentropic ascent (anomalous upward tilt toward the active phase) and diabatic
ascent (anomalous radiative heating where CPT temperatures are cold) may contribute, as may the changing boundaries of the
anticyclone itself, under the assumption that relatively humid air is mainly confined to the active core. However, we note
that deeper convective activity and more extensive high cloud cover are again located upstream of the water vapor maximum
(Fig. 12a; Figs. S3, S16), though this arrangement is more pronounced in PC3+ (the 'Iranian Plateau' mode) than in PC3−
(the 'Tibetan Plateau' mode). Over the eastern Tibetan Plateau, OLR anomalies are noticeably larger in ERA5 (Fig. 12a) than
in JRA-3Q or MERRA-2 (Fig. S16), which may help to explain the sharper southwest-to-northeast tilt of this mode in the
ECMWF reanalyses (i.e. CAMS and ERA5). Supporting this point, the inter-reanalysis discrepancy in the tilt of the dipole
is found only in water vapor and not in ozone or CO (Fig. 12c,d; Figs. S8g-i and S15), suggesting that it may be explained
by stronger convective injection in ERA5 and CAMS, which is then constrained by local dehydration in the cold anticyclonic
vortex.

As in PC2, anomalies in temperature, radiative heating, MSF, ozone, and CO (Fig. 12) are highly coherent in both poles of
the PC3 east–west dipole, demonstrating the dynamical reorganization of the anticyclone associated with PC3. PC3+ is linked
to anomalous convective activity over South Asia, where stronger heating induces a westward shift, while PC3− is linked to
anomalous convective activity over the western Pacific, where stronger heating induces an eastward shift (e.g. Nützel et al.,
2016, their Fig. 15). However, our results do not rule out a role for convective injection, with the reanalyses showing enhanced
deep convection over the Indus Valley and the Karakoram Gap when the anticyclone center is shifted westward and over the
Sichuan Basin when the anticyclone center is shifted eastward (Fig. 12a; Fig. S16). Such shifts in deep convection may help
to 'fill up' the meridionally-displaced anticyclone cores with freshly detrained air (e.g. Heath and Fuelberg, 2014), with the
caveat such moistening effects could be quickly damped by the negative temperature anomalies near the tropopause (Fig. 14c).

The reanalysis products agree well with each other and with Aura MLS in their representations of anomalies in dynamical
fields, temperature, cloud cover, ozone, and CO associated with PC2 and PC3 (Fig. S8; Figs.S12–S16). ERA5 shows the
largest shifts and deepest penetration of anomalous deep convection along 25°N, with changes in MERRA-2 and JRA-3Q
more confined to the upper troposphere, especially in PC2 (i.e. $p > 150$ hPa; Fig. S3). Therefore, although all three reanalyses
are consistent in the qualitative east–west shift of convective activity along this transect, we cannot confidently infer the
implications for convective injection above the LZRH.

To emphasize these points, Fig. 15 shows anomalous water vapor tendencies due to resolved moisture flux convergence
and parameterized physics ($\mathrm{PCWV_{dyn+phy}}$) during the transition phases between PC2 and PC3. Here, the transition phases
are defined as the three pentads ending on the corresponding phase of PC2 or PC3 with no lead–lag requirement, so that all
events are sampled and each sampled event includes 15 days. The centers of water vapor enhancement for each phase, where
fractional changes in PCWV exceed 10%, are delineated by pink contours. Despite some differences in magnitude, anomalies
in $\mathrm{PCWV_{dyn+phy}}$ from these three reanalyses are broadly consistent, indicating that the reanalyses produce these modes
primarily through internal processes. The one exception to this consistency is the western center of PC2 in JRA-3Q, where a




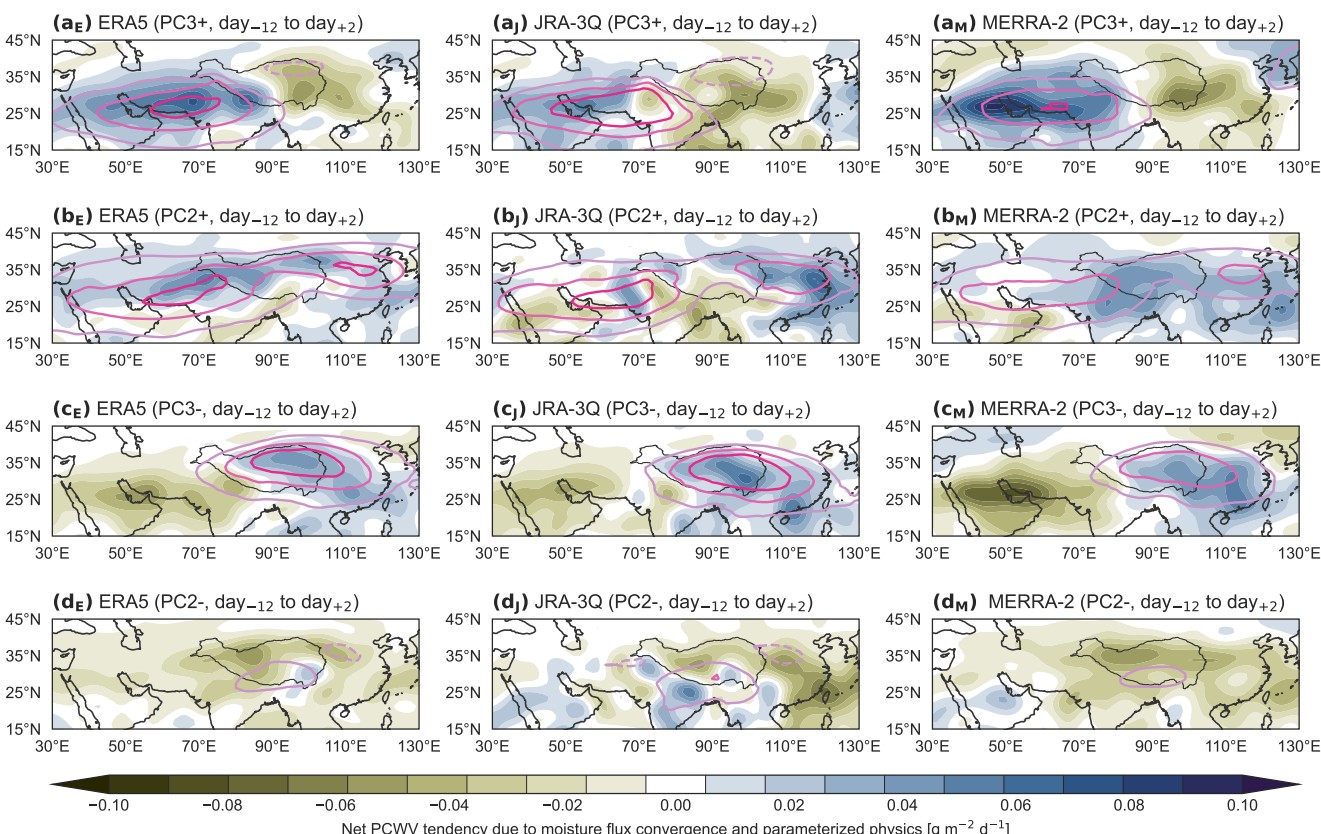

**Figure 15.** Anomalous water vapor tendencies due to dynamics (i.e. moisture flux convergence) and parameterized physics for ($a_E$–$d_E$) ERA5, ($a_J$–$d_J$) JRA-3Q, and ($a_M$–$d_M$) MERRA-2 during the 15 days ending on episodes of (a) positive PC3 > 1 (PC3+), (b) positive PC2 > 1 (PC2+), (c) negative PC3 < −1 (PC3−), and (d) negative PC2 < −1 (PC2−). Anomalies in partial column water vapor are marked with pink contours at intervals of 10% from ±10% (negative dashed) during each phase. The location of the Tibetan Plateau is marked by a dark grey contour in all panels.





weak dynamical term (Fig. S17) and a moderately strong physics term (Fig. S18) produce net tendencies that are largely out
of phase with ERA5 and MERRA-2. Discrepancies between JRA-3Q and the other reanalyses in the southwestern quadrant
are reduced by relatively large data assimilation increments there (Fig. S19). In all three reanalyses, the physics and dynamics
terms mainly compensate each other. These individual terms have magnitudes that exceed their sum by more than a factor 10,
and their magnitudes and the detailed distributions are more varied across the reanalyses (Figs. S17 and S18). Assimilation
increments are even more diverse (Fig. S19), with magnitudes only slightly less than the sum of the dynamics and physics
terms $PCWV_{dyn+phy}$.

## 4 Discussion

Our application of EOF analysis to water vapor fields emphasizes situations for which processes that influence water vapor
variability are aligned, as opposed to situations for which these influences counteract each other. For example, the east–west
dipole mode (Fig. 2b) shows 'Tibetan Plateau' and 'Iranian Plateau' phases linked to the propagation of convectively-coupled
Rossby waves across the southern part of the domain (Fig. 11; see also Garny and Randel, 2013; Ren et al., 2019). Although
this mode is consistently one of the leading modes of variability in water vapor in the data products considered here, it explains
only 3–6% of the variance in water vapor. For context, the corresponding east–west dipole modes based on Aura MLS ozone
and CO (not shown) explain more than 10% of the variance (2–3 times that for water vapor) and appear as PC2. This differ-
ence suggests that the water vapor mode results from two separate features: a shift in the anticyclone center and anomalous
convection that reaches deep enough to 'fill up' that shifted center out of phase with the CPT anomaly (Heath and Fuelberg,
2014). ERA5 anomalies in cloud ice and convective heating associated with this mode (Fig. 12a) are consistent with this idea:
anomalous cloud ice and heating approach 100 hPa in the southwestern part of the domain over the Arabian Sea in the moist
(Iranian Plateau) phase and over the southeastern Tibetan Plateau and Sichuan Basin in the dry (Tibetan Plateau) mode. PC2
shows greater consistency across results based on Aura MLS water vapor, ozone, and CO, indicating a greater overlap of the
convective, dynamical, and temperature anomalies that favor a humid tropopause layer. In particular, water vapor anomalies
associated with PC2 are in phase rather than out of phase with anomalies in CPT temperatures.

Many studies have explored the mechanisms behind convection, composition, and circulation covariability in the UTLS
above the ASM. Among these, Garny and Randel (2013) emphasized the control exerted by convective heating on the ASM
anticyclone on subseasonal timescales, highlighting how PV anomalies derive largely from shifts in large-scale convective
heating between the maritime monsoon and the South Asian subcontinent. Their results, which link negative PV anomalies (a
locally enhanced anticyclone) with positive CO anomalies, echo our results for PC2 and PC3 (section 3.3). Nützel et al. (2016)
and Manney et al. (2021) evaluated reanalysis representations of the ASM anticyclone in pressure and potential temperature
vertical coordinates, respectively. Manney et al. (2021) further assessed internal variability, finding significant correlations
between anticyclone area and ENSO but little relationship with the QBO after June. However, they only calculated correlations
within ±2 months lag, and the correlations reported for ENSO were weak at the altitudes we consider here. Influences of ENSO
on the anticyclone were also reported by Nützel et al. (2016) and Yan et al. (2018), among others, but these studies focused




more on how ENSO regulates subseasonal variability. Manney et al. (2021) found that the area and duration of the anticyclone increased significantly over the period 1979–2018, while Zolghadrshojaee et al. (2024) recently reported long-term increases in tropical CPT temperatures. These changes may help to explain the strong reanalysis-based moistening trends and/or the absence of such a trend in the Aura MLS record; however, detailed analysis is deferred to future work.


Randel et al. (2015) focused more specifically on the mechanisms behind water vapor variability at 100 hPa, highlighting the preeminent role of temperature anomalies in determining water vapor at this level. Our results are broadly consistent with this interpretation, with water vapor anomalies in phase with CPT temperature anomalies in both PC1 and PC2. However, water vapor anomalies are out of phase with CPT temperature in PC3 (and PC1$_{\mathrm{TREND}}$ based on reanalysis products) and

the largest convective and water vapor anomalies in PC2 are located downstream of anomalously warm CPT temperatures. Our results therefore suggest that deep convective sources may play a larger role in regulating water vapor in this region than that concluded by Randel et al. (2015). Satheesh Chandran and Sunilkumar (2024) analyzed how quasi-biweekly variations in monsoon convection impact composition and chemical processing within the monsoon anticyclone. Their results are mostly consistent with our PC2 and PC3 (section 3.3), but with some distinctions that further highlight the importance of anomalously

deep convection in modulating the signature of quasi-biweekly waves in UTLS water vapor. In particular, they reported a weaker signal in water vapor associated with the Tibetan Plateau phase, suggesting that convective anomalies under the shifted anticyclone center may not be deep enough to overcome out-of-phase CPT temperature anomalies. QBO influences on water vapor in the ASM UTLS were recently evaluated by Peña Ortiz et al. (2024). However, their work focused on the QBO influence on quasi-biweekly waves and distinct imprints of QBO on convective heating and transport to the tropopause between July and

August. By contrast, our results highlight the QBO influence on interannual variability (section 3.2), mainly through anomalous subsidence and warming on the tropical flank of the anticyclone as the QBO westerly phase reaches the tropopause.

At the subseasonal scale, we highlight the influences of quasi-biweekly waves (see also Randel and Park, 2006; Ren et al., 2019). Quasi-biweekly waves in the Asian monsoon system have been argued to arise from instabilities in the monsoon flow downstream of anomalous heating over the Tibetan Plateau (Liu et al., 2007). However, the sequence of large-scale organized

convection in these waves is often viewed as starting from the western North Pacific east of the Philippines and then propagating northwestward (e.g. Kikuchi and Wang, 2009; Wang and Chen, 2017). During most of this propagation, these waves have limited impact near the tropopause due to the buffering effect of the upper-level easterly jet along the southern flank of the anticyclone. The convective anomaly deepens and intensifies when it reaches the Indus River Valley, where the buffering effect is much weaker. The heating associated with this convection triggers an upper-level high that disturbs the subtropical

jet (Fig. 11d-ii). Downstream restoring of that disturbance produces a second high over northern China (Fig. 11a-i,a-ii), which propagates southward in the eastern flank of the anticyclone before breaking over eastern China or the western North Pacific (Fig. 11b-i,b-ii; Ortega et al., 2017). Strong upper-level wave activity fluxes along the eastern flank of the anticyclone are evident as first the anticyclonic anomaly and then a following cyclonic anomaly propagate southward after PC2+ (Fig. 11b-ii,c-i) and PC3− (Fig. 11c-ii,d-i), respectively. The former stage corresponds to the emergence of a convective anomaly over

the western North Pacific and the latter stage to the arrival of this convective anomaly over southern China (Fig. 11c-i,c-ii). The southward fluxes shift toward the west as the convection center begins to emerge over the Arabian Sea and adjacent coastal



areas (Fig. 11d-ii,a-i), and then become predominantly eastward as the upper-level high retracts into the subtropical jet waveguide (Fig. 11a-ii,b-i). Our results therefore support the contention of Ortega et al. (2017) that the upper-level expression of quasi-biweekly waves in the Asian monsoon system is not a passive response to the anomalous convective heating, but part
of a coupled pattern that cycles around the anticyclone. This cycle, recently been examined in more detail by Amemiya and Sato (2018, 2020), represents an important and underexplored connection between variability in the upper-level anticyclone and surface weather in the monsoon region at subseasonal scales.

## 5    Conclusions

We have used pentad-resolution gridded data from Aura MLS satellite observations and five atmospheric reanalyses (MERRA-
2, M2-SCREAM, CAMS, ERA5, and JRA-3Q) to examine variations in the horizontal and vertical distribution of water vapor in the tropopause layer (147 hPa–68 hPa) above the Asian summer monsoon (ASM). We have further evaluated the mechanisms behind these variations during the warm seasons (May—September) of 2005 through 2021 by evaluating covariability with ozone, carbon monoxide (CO), and dynamical and thermodynamic fields, including detailed water vapor budgets that explicitly separate the effects of resolved transport, parameterized physics, and data assimilation. The results serve three purposes:
reanalysis assessment, reanalysis intercomparison, and mechanistic analysis.

By applying principal component analysis to the vertical and horizontal variability of deseasonalized anomalies, we have identified and described three modes of spatio-temporal variability in water vapor above the ASM. Comparing five reanalyses and observational products, we identify a remarkable consistency in UTLS water vapor (for which increments from assimilated observational data are weak and sometimes detrimental; Davis et al., 2017; Kosaka et al., 2024), the terms of the tropopause-
layer moisture budget, and covariations in dynamical and thermodynamic fields. This overall consistency indicates that the current generation of reanalysis products is increasingly able to reproduce the characteristics and mechanisms of water vapor variability in this part of the atmosphere. The three modes we analyze correspond to key signatures of variability in UTLS humidity in this region from intraseasonal to interannual scales. The interannual mode, which features regional-scale moist or dry anomalies, is strongly correlated with the quasi-biennial oscillation (QBO). The phase of this mode is usually set before
the monsoon and then maintained as the monsoon evolves. The second and third modes highlight different phases of quasi-biweekly waves, as convective anomalies propagate from the maritime monsoon region in the southeast through Southeast Asia, Bangladesh, and North India to the Indus Valley and Karakoram Gap in the northwest (see also Randel and Park, 2006; Nützel et al., 2016; Ortega et al., 2017; Ren et al., 2019; Satheesh Chandran and Sunilkumar, 2024). Although these modes often occur in quadrature, with PC2 following PC3 and the opposite phase of PC3 following PC2, each at roughly 10 days lag,
the lead–lag correlations are relatively small (peak $|r| \approx 0.2$–$0.3$), suggesting that the influences of other modes of organized convective variability on water vapor in the tropopause layer also project onto these patterns.

In contrast to the good overall agreement between Aura MLS and the reanalysis products in detrended variability, linear trends in tropopause layer water vapor over 2005–2021 differ in both magnitude and sign over much of the region, especially below the tropopause. Although the reanalyses are remarkably consistent with each other, our analysis provides several reasons



to doubt the reanalysis-based trends. First, linear trends in ozone and CO also show discrepancies between Aura MLS and the reanalyses. Second, trends in cold point tropopause (CPT) temperatures are negative in the southeastern quadrant of the anticyclone where the reanalyses show the largest positive trends in water vapor. Trends toward colder temperatures imply drier conditions near the tropopause, as observed by Aura MLS but not reproduced by the reanalyses. Further analysis of the tropopause-layer water vapor budgets for ERA5, JRA-3Q, and MERRA-2 show that the reanalysis trends are driven by

processes outside the monsoon domain and season. As a consequence, trends based on Aura MLS and the reanalyses are in best agreement above the CPT near 68 hPa, where air masses are less affected by transport from the monsoon circulation below and less confined within the boundaries of the anticyclone (Legras and Bucci, 2020; Nützel et al., 2019). Further investigation will be needed to identify the reasons behind this discrepancy.

The influences on composition in the monsoon anticyclone reported in this work may extend into the extratropical and

tropical lower stratosphere, as Rossby wave breaking and eddy shedding events transmit the characteristics of air in this region far beyond the seasonal and geographic boundaries of the monsoon (e.g. Dethof et al., 1999; Wright et al., 2011; Ploeger et al., 2013; Yan et al., 2019). Future work may design transport simulations to target the composite evolution of convectively detrained air under these different modes, including typical transitions among the phases of the subseasonal modes. Such simulations could leverage increased consistency in reanalysis representations of deep convective and radiative heating in the

monsoon tropopause layer to better quantify the influences of monsoon convection in different regions, assess key uncertainties, and estimate the potential impacts of future climate change in this system.

*Data availability.*  Aura MLS (Lambert et al., 2021; Schwartz et al., 2021a, b), MERRA-2 (GMAO, 2015a, b, c, d), and M2-SCREAM (GMAO, 2022) data were acquired from the NASA Goddard Earth Sciences Data and Information Services Center (GES DISC; https://disc.gsfc.nasa.gov). ERA5 products (Hersbach et al., 2017, 2023a, b) were acquired from the Copernicus Climate Data Store (CDS; https://cds.climate.copernicus.eu)

and CAMS reanalysis products from the Copernicus Atmosphere Data Store (ADS; https://ads.atmosphere.copernicus.eu/cdsapp#!/dataset/cams-global-reanalysis-eac4). JRA-3Q reanalysis products (JMA, 2022) were acquired from the Data Integration and Analysis System (DIAS; https://search.diasjp.net/en/dataset/JRA3Q) archive maintained by the Japan Agency for Marine-Earth Science and Technology (JAMSTEC) and the University of Tokyo. The Berlin Quasi-Biennial Oscillation time series has been acquired from the updated archive maintained by the Karlsruhe Institute of Technology (https://www.atmohub.kit.edu/english/807.php). The Oceanic Niño Index time series was acquired from

the National Oceanic and Atmospheric Administration (NOAA) Climate Prediction Center (https://www.cpc.ncep.noaa.gov/data/indices/) index calculated with centered base periods using version 5 of the Extended Reconstructed Sea Surface Temperature (National Centers for Environmental Information, NESDIS, NOAA, U.S. Department of Commerce, 2019).

*Author contributions.*  SZ, JC, and JSW conceived the study and wrote the initial draft of the paper. SZ, JC, JSW, JG, and NZ conducted the analysis. XY, PK, ML, SMD, and ST provided feedback on the analysis and assisted with interpretation. GJZ, JSW, and PK acquired funding

for the work. All authors contributed to writing and revising the manuscript.




*Competing interests.* One of the authors is a guest coordinator for the "SPARC Reanalysis Intercomparison Project (S-RIP) Phase 2" special issue in Atmospheric Chemistry and Physics. The authors have no other competing interests to declare.

*Acknowledgements.* We thank Dr Kris Wargan for assistance confirming details of the MERRA-2 and M2-SCREAM reanalyses, Dr Yayoi Harada for assistance confirming details of the JRA-3Q reanalysis, and Dr Hans Hersbach, Dr Antje Inness, and Dr Peter Bechtold for assistance confirming details of the ERA5 and CAMS reanalyses. This work has been supported by the National Natural Science Foundation of China (grant number 42275053), the Beijing Municipal Natural Science Foundation (grant number IS23121), and the Hong Kong Research Grants Council (project no.16300424).




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
