# Peer review of "Covariability of dynamics and composition in the Asian monsoon tropopause layer from satellite observations and reanalysis products"

_EGUsphere, 2025_

## Author Comment (AC2)

**Author Reply to Reviewers**

We thank Dr. Mengchu Tao and Reviewer 2 for raising a number of interesting questions. Addressing these questions has already helped tremendously to tighten the analysis and make it more incisive, as described below. The resubmission will be much stronger for your efforts. Initial responses to all general comments and selected technical comments are provided below.

**1 Response to reviewer 1**

**Reviewer 1 evaluations:** This paper focuses on analyzing water vapor variability in the Upper Troposphere–Lower Stratosphere (UTLS) region of the Asian Summer Monsoon (ASM) area. The authors use satellite data from Aura MLS and five atmospheric reanalysis datasets, including MERRA-2, M2-SCREAM, CAMS, ERA5, and JRA-3Q, to conduct a spatiotemporal mode analysis. The main results reveal three key modes of variability, including PC1, large-scale regional water vapor wetting or drying anomalies, PC2 and PC3, intraseasonal oscillations linked to quasi-biweekly variability.

A key conclusion in the paper, from my point of view, is the discrepancies between reanalysis products and Aura MLS data regarding spatial distribution and sign (positive/negative) of water vapor trends. While most reanalysis products show an increasing water vapor trend in the southeast of Asian monsoon during warm season, their spatial characteristics differ significantly from those derived from Aura MLS.

Overall, the paper is novel in design, well-written, and thorough in its data analysis, comparisons of methods, and results. I recommend minor revisions to address the following points for further improvement.

**1.1 General comments**

> **Comment 1.1**
>
> The analyses, particularly those focused on interannual variability and intraseasonal oscillations, are convincing and well-executed. My concerns are mainly about the analysis of PC1 trends. While the discussion clearly identifies the differences in trends between Aura MLS and reanalysis products, I would suggest a deeper exploration of the reasons behind these discrepancies.

**Response:**

Thank you for this suggestion. We have conducted further analysis and incorporated new results accordingly.

As documented by Wright et al., 2025 (see their Figure 1), the cold point tropopause (CPT) during our analysis period is located between 85–95 hPa. In the following, we adopt this estimate of the CPT pressure as the threshold to distinguish the upper troposphere (UT) from the lower stratosphere (LS). The partial column water vapor (PCWV) above the tropopause is defined as the water vapor integrated from 83 to 68 hPa, whereas PCWV below the tropopause is integrated from 147 to 100 hPa. Figure R1 presents the deseasonalized anomalies of PCWV above the tropopause regressed onto the first principal component (PC1), its trend ($PC1_{TREND}$), and interannual variability ($PC1_{IAV}$). Conversely, Figure R2 shows the deseasonalized anomalies of PCWV below the tropopause, regressed onto the PC1 interannual variability and the second principal component (PC2). Our analysis focuses on the reanalysis datasets from M2-SCREAM, JRA-3Q, CAMS, and ERA5, excluding MERRA-2 due to its inability to represent variability in the LS.

The spatial distribution of PCWV (Fig. R1) regressed onto PC1 above the tropopause (hereafter, $PC1^{strat}$) aligns well with the spatial pattern regressed onto PC1 in the UTLS (i.e., Fig. 1 in the manuscript). Meanwhile, the spatial distributions regressed onto PC1 and PC2 below the tropopause (hereafter, $PC1^{trop}$ and $PC2^{trop}$, Fig. R2) correspond to anomalies regressed onto UTLS PC2 and PC3, respectively (i.e., Fig. 2 in the manuscript). This indicates that the PC signals identified within the UTLS (shown in the manuscript) encompass both the LS (represented by PC1 in the manuscript) and the UT (represented by PC2 and PC3 in the manuscript). Specifically, PC1 (in the manuscript) captures interannual variability in the lower stratosphere ($PC1^{strat}$), above the tropopause, whereas PC2 and PC3 (in the manuscript) reflect subseasonal variability in the upper troposphere ($PC1^{trop}$ and $PC2^{trop}$), below the tropopause. The high correlations observed between the UTLS principal components and those above/below the tropopause further support this interpretation (Fig. R3). Consequently, the discrepancies in trend between Aura MLS and reanalysis products predominantly stem from different trends in water vapor in the lower part of the tropopause layer (i.e., 147 hPa to 100 hPa). All reanalysis products simulate a regional moistening trend in water vapor above the tropopause, consistent with Aura MLS (Fig. R1b,e,h,k,n).

[Figure]

Figure R1: Deseasonalized partial-column water vapor (PCWV) anomalies integrated from 83 to 68 hPa regressed onto the (a) first principal component (PC1) from Aura MLS and its (b) trend (PC1$_{\mathrm{TREND}}$) and (c) interannual variability (PC1$_{\mathrm{IAV}}$) components; (d)–(f) same as (a)–(c), but for PC1 from M2-SCREAM; (g)–(i) same as (a)–(c), but for PC1 from JRA-3Q; (j)–(l) same as (a)–(c), but for PC1 from CAMS; (m)–(o) same as (a)–(c), but for PC1 from ERA5; and (p)–(r) the corresponding principal component time series. Principal components (PCs) are based on EOF analysis of vertical and horizontal variations in water vapor for the two Aura MLS pressure levels within 68 hPa–83 hPa, 30°E–130°E, and 15°N–45°N. Red contours mark the location of the Tibetan Plateau, with stippling indicating significance at the 95% confidence level based on Student's $t$ test. The fraction of total variance explained by each mode is listed at the upper right of panels(a,d,g,j,m). Correlations between MLS-based PCs and those based on M2-SCREAM (light red), JRA-3Q (purple), CAMS (light blue), and ERA5 (dark blue) are listed from right to left along the tops of panels (p)–(r).

[Figure]

Figure R2: Deseasonalized partial-column water vapor (PCWV) anomalies integrated from 147 to 100 hPa regressed onto the (a) first principal component (PC1) interannual variability (PC1$_{IAV}$) and (b) the second principal component (PC2) from Aura MLS; (c)–(d) same as (a)–(b), but for PC1$_{IAV}$ and PC2 from M2-SCREAM; (e)–(f) same as (a)–(b), but for PC1$_{IAV}$ and PC2 from JRA-3Q; (g)–(h) same as (a)–(b), but for PC1$_{IAV}$ and PC2 from CAMS; (i)–(j) same as (a)–(b), but for PC1$_{IAV}$ and PC2 from ERA5; and (k)–(l) the corresponding principal component time series. Principal components (PCs) are based on EOF analysis of vertical and horizontal variations in water vapor for the three Aura MLS pressure levels within 100 hPa–147 hPa, 30°E–130°E, and 15°N–45°N. Red contours mark the location of the Tibetan Plateau, with stippling indicating significance at the 95% confidence level based on Student's $t$ test. The fraction of total variance explained by each mode is listed at the upper right of panels(a–j). Correlations between MLS-based PCs and those based on M2-SCREAM (light red), JRA-3Q (purple), CAMS (light blue), and ERA5 (dark blue) are listed from right to left along the tops of panels (k)–(l).

[Figure]

Figure R3: Principal component (PC) time series for the spatial patterns shown in Figs. R1 and R2 based on Aura MLS (black), ERA5 (dark blue), CAMS (light blue), JRA-3Q (purple), and M2-SCREAM (light red) and PCs shown in the manuscript based on Aura MLS (dashed black). Time series are shown for (a) the first principal component (PC1) interannual variability in the UTLS ($PC1_{IAV}$) and above the tropopause ($PC1_{IAV}^{strat}$); (b) the second principal component (PC2) in the UTLS and PC1 interannual variability below the tropopause ($PC1_{IAV}^{trop}$); and (c) the third principal component (PC3) in the UTLS and PC2 below the tropopause ($PC2^{trop}$). PCs in the UTLS are based on EOF analysis of vertical and horizontal variations in water vapor for the five Aura MLS pressure levels within 68 hPa–147 hPa, 30°E–130°E, and 15°N–45°N; PCs above the tropopause are based on EOF analysis of vertical and horizontal variations in water vapor for the three Aura MLS pressure levels within 68 hPa–83 hPa, 30°E–130°E, and 15°N–45°N; and PCs below the tropopause are based on EOF analysis of vertical and horizontal variations in water vapor for the three Aura MLS pressure levels within 100 hPa–147 hPa, 30°E–130°E, and 15°N–45°N. Correlations between MLS-based PCs in the UTLS and those based on MLS above/below the tropopause (black), M2-SCREAM (light red), JRA-3Q (purple), CAMS (light blue), and ERA5 (dark blue) are listed from right to left along the tops of panels (a–c).

[Figure]

Figure R4: Deseasonalized partial-column water vapor (PCWV) anomalies integrated (a) from 147 to 68 hPa and (b) from 147 to 100 hPa regressed onto the first principal component (PC1) trend variability (PC1$_{\text{TREND}}$) from Aura MLS (Fig. R1q); (c)–(d) same as (a)–(b), but for PCWV from M2-SCREAM; (e)–(f) same as (a)–(b), but for PCWV from JRA-3Q; (g)–(h) same as (a)–(b), but for PCWV from CAMS; (i)–(j) same as (a)–(b), but for PCWV from ERA5. Principal components (PCs) are based on EOF analysis of vertical and horizontal variations in water vapor for the two Aura MLS pressure levels within 68 hPa–83 hPa, 30°E–130°E, and 15°N–45°N. Red contours mark the location of the Tibetan Plateau, with stippling indicating significance at the 95% confidence level based on Student's $t$ test.

**Comment 1.2**

Another question from my side, whether the second reason to doubt the reanalysis-based trends are robust: "trends in cold point tropopause (CPT) temperatures are negative in the southeastern quadrant of the anticyclone where the reanalyses show the largest positive trends in water vapor" (in conclusion). If reanalysis WV increase mainly under the tropopause and thus increase the PCWV, it seems be consistent with OLR/cloud trend (convection increase pattern shown in Figure 6). It thus meets "criteria 3: a plausible physical mechanism". And WV decrease due to local CPT decrease is not a main driver since the mass of WV in the LS is much less than that in the UT. Surely, long-term warming of the tropical cold point tropopause increasing WV outside the monsoon region can be another reason. But I don't see the contradiction between CPT cooling and PCWV increasing over one region.

To shed more lights on this point, my suggestion could be:

1) the trends (and their spatial characteristics) be further decomposed into contributions above and below the tropopause;

2) further analysis of the "dyn" and "phy" terms individually (specifically their behaviours above and below the tropopause) in Fig. 7. This could potentially reveal whether convection is the primary driving factor in the trends observed. And this could uncover systematic patterns or consistencies within these two terms from reanalysis datasets.

**Response:**

Thank you for your detailed and insightful comment. We agree that the trend below the tropopause meets the third criterion (but not the second) and will work to make this clearer in the revised text. This possibility motivates our call for a more complete analysis of trends in UTLS composition and related fields in this region at the end of section 3.1.

As discussed in our response to Comment 1.1, PC1 in our manuscript primarily corresponds to interannual variability in the humidity of the lower stratosphere (PC1$^{\mathrm{strat}}$, see Fig. R1 and Fig. R3a). Both reanalysis products and observations indicate a regional moistening trend above the tropopause (Fig. R1, R5a). Although these moistening trends occur downstream of the positive anomalies in the cold point tropopause (CPT) (Fig. R5), consistent with the expectation that a warmer CPT should enhance water vapor content above the cold point, it is difficult to relate these to the enhanced convection in the southeast quadrant. The difficulty arises because there is a trend toward colder cold point temperatures co-located with that increase in convective activity. To evaluate this in more detail, we will examine the changes in tendency terms between these two regions. For example, if the positive trend in the CPT and the positive trend in LSWV are mechanistically related, then we may expect to see a positive trend in the physics term (less condensation drying) in the locations where the CPT has warmed, all else remaining equal.

In an area- and annual-mean sense, trends in resolved transport (DYN; $-\nabla \cdot (\mathbf{V}q) - \partial(\omega q)/\partial p$) are small but positive both above (Fig. R6) and below the tropopause (Fig. R7) in the reanalyses. However, the increases in these transports are offset by a decreasing trend in the physical tendencies ($S_{\mathrm{phy}}$), and sometimes more than offset, as is the case in MERRA-2. The net changes in PHY + DYN trends are inconsistent across different products and are effectively offset by the data assimilation components ($S_{\mathrm{ana}}$), regardless of the altitude relative to the tropopause. We interpret this lack of any significant trend in the ASM UTLS budget as indicating that the consistent moistening trend in LS water vapor is driven by processes outside the monsoon season or domain. Because the trend in the upper troposphere is more spatially heterogeneous, analysis of the area-mean budget is insufficient. We will investigate the mechanisms in the trend below the tropopause as we prepare the revised manuscript and include an update when we submit that revision.

**1.2 Specific comments**

**Comment 1.3**

Following EQ (2) in Line 160, the terms "Sphy" and "Sana" should be briefly explained. Specifically: What key physical processes related to water vapor are captured by "Sphy" (e.g., condensation, deposition, subsidence, etc.)? What is the role of "Sana," particularly in relation to the data assimilation process? I also wonder whether subgrid-scale mixing is included in the "Sres" term?

**Response:**

Thank you for this suggestion. We will explain these terms in more detail in the revised manuscript.

$S_{\mathrm{phy}}$ comprises the influences of parameterized physical processes, including cloud microphysics, convection, and turbulent mixing (see Wright et al. (2025), their Fig. 8). Subgrid-scale mixing is included in $S_{\mathrm{phy}}$. $S_{\mathrm{res}}$ is a 'diffusive' residual, which we interpret this residual as primarily representing the transport due to high-frequency or small spatial scale motions resolved by the reanalysis model but not by our calculation of $S_{\mathrm{dyn}}$, together with the effects of numerical diffusion. To reiterate, we calculate $S_{\mathrm{dyn}}$ on a coarser spatial grid (for ERA5, by a factor 4) and the analysis interval (1 h) is five times

[Figure]

Figure R5: (a) Deseasonalized partial-column water vapor (PCWV) anomalies integrated from 83 to 68 hPa regressed onto the first principal component (PC1) trend variability (PC1$_{\text{TREND}}$) from Aura MLS. Changes in cold point tropopause (CPT) temperatures based on (b) ERA5, (c) MERRA-2, and (d) JRA-3Q regressed onto PC1$_{\text{TREND}}$. The location of the Tibetan Plateau is marked by a red contour in all panels. Stippling indicates locations where regression slopes are significant at the 95% confidence level based on Student's $t$ test.

[Figure]

Figure R6: Yearly variations in the sum of the dynamics and physics terms (green lines), assimilation increments (yellow lines), dynamics terms (brown lines), physics increments (red lines), and time rate of changes in partial column water vapor (gray boxes) above the monsoon tropopause layer (68-83 hPa) based on (a,b) ERA5, (c,d) JRA-3Q, and (e,f) MERRA-2 over (a,c,e) the southwestern quadrant (15°N–30°N, 30°E–80°E) and (b,d,f) the southeastern quadrant (15°N–30°N, 80°E–130°E) of the monsoon anticyclone. Correlation coefficients between net water vapor tendency time series based on individual reanalyses are listed in the lower left corner of each panel.

[Figure]

Figure R7: Yearly variations in the sum of the dynamics and physics terms (green lines), assimilation increments (yellow lines), dynamics terms (brown lines), physics increments (red lines), and time rate of changes in partial column water vapor (gray boxes) below the monsoon tropopause layer (100–147 hPa) based on (a,b) ERA5, (c,d) JRA-3Q, and (e,f) MERRA-2 over (a,c,e) the southwestern quadrant (15°N–30°N, 30°E–80°E) and (b,d,f) the southeastern quadrant (15°N–30°N, 80°E–130°E) of the monsoon anticyclone. Correlation coefficients between net water vapor tendency time series based on individual reanalyses are listed in the lower left corner of each panel.

longer than the model time step (every 12 minutes). As a result, our calculation is missing a chunk of the transport that the model resolves. Because the model resolves this transport, it does not appear in the physics term. Because our calculation of moisture flux divergence does not resolve this transport, it does not appear in the dynamics term. We have struggled to come up with a good name for this term that clearly conveys what it represents ('resolved mixing' as opposed to 'subgrid-scale mixing'? 'eddy transports'?) and would welcome suggestions. Finally, the residual term also includes the effects of numerical diffusion.

$S_{ana}$ is the data assimilation term. MERRA-2 explicitly provides $S_{ana}$, but this component must be estimated for budgets based on ERA5 and JRA-3Q. We estimate $S_{ana}$ for ERA5 and JRA-3Q by directly subtracting forecast specific humidities (the model-generated background state before data assimilation) from analysis specific humidities (the final reanalysis product after data assimilation). We then average this difference and multiply by the number of analysis cycles per day (two for ERA5 and four for JRA-3Q) to get an assimilation-related moistening rate in units of per day (alternatively we could divide by the length of the analysis cycle). More details on these terms and their interpretation have been provided by Wright et al. (2025) in their sections 2.1 and 4.

**Comment 1.4**

Figure 1: The titles of the three subplots for MERRA-2 (panels am-cm) are incorrect; they should refer to "PC1" instead of "PC2."

**Response:**

Thank you for your careful check. However, due to the distinct characteristics of different reanalysis datasets in the upper troposphere and lower stratosphere (UTLS), the principal components (PCs) derived from each dataset are not necessarily directly comparable on a one-to-one basis. We therefore attempt to match the modes in terms of their spatial signatures. Under this approach, PC2 in MERRA-2 corresponds to PC1 in other reanalysis datasets. The reason for this difference is that MERRA-2 relaxes stratospheric water vapor to a zonal-mean climatology with a 3-day relaxation time scale, washing out low-frequency variability in the lower stratosphere. PC1 based on Aura MLS is defined largely by the variance in lower stratospheric water vapor, so MERRA-2 does not represent this mode well.

> **Comment 1.5**
>
> Figure 3 (panel b): Please reduce the y-axis range to between -2 and 2 to allow for better visualization. Consider setting the y = 0 reference line to gray for improved clarity.

**Response:**

We appreciate your suggestion. The y-axes of all panels in the figure were set to a fixed range of -4 to 4 to ensure facilitate comparison across the panels. However, we acknowledge that this choice makes it more difficult to compare the line profiles in panel b effectively and will revise Figure 3 to address this issue.

> **Comment 1.6**
>
> I wonder why MLS trend show larger positive trend than merra-2 (Fig. 3)? It seems conflict with Fig.1. Does that mean MLS PC1 trend for the whole region is positive? And the positive trend is highly contributable from 35-45N latitude band according to Fig.1 (b)?

**Response:**

Thank you for raising this question, which highlights a logic error in our presentation of Fig. 1. MERRA-2 exhibits the smallest trend in the low-frequency PC among the datasets we examine, corresponding to $PC2_{TREND}$ in MERRA-2. This is primarily because MERRA-2 damps lower stratospheric water vapor variability through relaxation toward a zonal-mean annually-repeating seasonal cycle, resulting in comparatively small year-to-year variability and trend signals. Because the trend in the lower stratosphere (the main point of consistency between Aura MLS and the reanalyses) is small, the low-frequency mode in MERRA-2 is dominated by variations below the tropopause and tilts more toward $PC2_{IAV}$. Conversely, MERRA-2 shows larger PCWV anomalies when regressed onto the trend component compared to MLS. This is because all regressions were shown for $PC_{TREND} = 1$. This approach is appropriate when the principal components are standardized, as in Fig. 2 and column (a) of Fig. 1. However, because we have not re-standardized $PC1_{TREND}$ and $PC1_{IAV}$ the magnitude of the water vapor anomalies associated with both modes is overstated in columns (b) and (c) of Fig. 1. This overstated amplitude is more pronounced for MERRA-2, because time variance in $PC2_{TREND}$ based on MERRA-2 is smaller than that for $PC1_{TREND}$ based on the other datasets. To correct this, we will keep the current time series of the trend and IAV parts of the low-frequency principal component but show regressions in columns (b) and (c) and Fig. 1 for the re-scaled trend and IAV time series.

Regarding your second question, yes, the positive trend in the full 147–68 hPa partial column is most robust along the northern edge of our analysis domain. However, Figs. R1 and R4 provide further insight on these spatial patterns as discussed in our response to comment 1.1 above. In our revised submission, we plan to use the differences in low-frequency modes for partial columns above and below the tropopause to clarify these details.

**2 Response to reviewer 2**

**Summary evaluation:** This work focuses on three leading modes of interannual variability in water vapor in the tropopause layer using the measurements from the MLS satellite and multiple reanalysis products. The first mode is linear trend and interannual variability in regional-scale anomalies, which show some differences in the satellite data and reanalysis. The second and third modes are related to anomalies within the monsoon anticyclone, such as, variabilities within the quadrants and a horizontal east-to-west dipole structure. The results show reanalysis captures the modes of variability in water vapor in the upper troposphere and lower stratosphere and the physical processes controlling them. The results shown here are based on comprehensive, thorough and very detailed analyses. However, I did not see clear motivation and goal of the work. Below are my comments for the authors might take into consideration.

**2.1 General comments**

**Comment 2.1**

L1 (Abstract): Rather than starting with 'we describe', I recommend start with some background information including why the Asian summer monsoon is important and what the goal of this study is. This will make the abstract more appealing.

**Response:**

Thank you for this suggestion. We removed this background information from the abstract at submission due to word count requirements, but will work to formulate a more appealing abstract that also meets the length requirement in the revised manuscript.

**Comment 2.2**

L16 (Introduction): It was mentioned that the goal of this study is to provide further insight into the mechanisms governing variations in water vapor. More specific information could be added here. What are examples of the mechanisms that we need to understand further? What variations in water vapor is discussed here? What is the science goal? A clear motivation and some scientific context of this work will be necessary. This work maybe relevant to the fact that there will be less observations of stratospheric water vapor available from satellites in the near future.

**Response:**

Thank you for raising these questions, which provide helpful guidance for us to present the work in a way that will be clearer and easier for readers to understand.

The research gaps and processes that we target in this work are articulated at the end of the first paragraph in the introduction: "Despite much progress in recent years, uncertainties remain regarding the relative influences and interplays among convective transport, the large-scale circulation, and thermodynamic structure near the tropopause in controlling humidity and composition in the upper troposphere and lower stratosphere (UTLS) above the ASM." We revisit this in the closing paragraph of the introduction, in the sentence following the one you reference: "We pay particular attention to characteristic patterns of covariability among convective activity, circulation patterns, and trace gas concentrations [as represented in current reanalysis systems]". To this, we should add that we evaluate the roles of data assimilation relative to those of parameterized physics and large-scale dynamics in allowing reanalysis systems to reproduce recurrent patterns of water vapor variability in the UTLS above the ASM.

**Comment 2.3**

In depth and very detailed analyses and descriptions of the results are presented in this work. I found it rather hard to understand all the detailed descriptions of figures. Many of the sentences are long and the description of results contains some speculation, besides facts. It would be helpful if some of the long sentences are split into multiple short sentences and simplify the descriptions.

**Response:**

We appreciate your suggestion and will make changes to the text accordingly.

**Comment 2.4**

It would also be helpful to include some context of the results from this work relative to previous studies throughout the main text. How is the result shown here different from previous work? Are they consistent with or different from previous work? This will help understand the results more scientifically. Is EOF analyses giving us new information that has not been discovered?

**Response:**

Thank you for this suggestion. Although direct comparisons are not always possible due to methodological differences, we have provided a discussion of our results in the context of previous work in section 4. It is also worth noting that the EOF analysis is primarily an exploratory tool. The first question we address with this tool is: are recurrent patterns similar between high-quality observations and reanalysis products? Having established that they are largely similar and delineated areas of disagreement, we address a second question: do these recurrent patterns derive from clear physical and dynamical mechanisms?

The decision to use EOF decomposition in this work was motivated by both the widespread use of this procedure in climate science and the platform it provides for us to identify and analyze recurrent patterns of water vapor anomalies. The EOF approach has known limitations (non-stationary, unable to cleanly identify propagating patterns) and many other statistical tools could provide a similar platform. Moreover, EOF patterns in isolation can be difficult to interpret and are often not physically meaningful. Although the use of vertically resolved information in identifying these patterns has not to our knowledge been applied to this region before, it is not the EOF analysis itself that provides new information, but rather our detailed decomposition of how the reanalyses generate the recurrent anomaly patterns identified by the EOF decomposition.

We will work to clarify these distinctions in the revised manuscript.

**Comment 2.5**

I think it would be helpful to provide some outlook. For instance, information about which reanalysis products represent dynamical or thermodynamical processes near the monsoon region well so that we can trust?

**Response:**

Thank you for this suggestion. We will collect and collate this information from the manuscript and include a summary of outlook recommendations in the final section.

**2.2 Specific Comments**

**Comment 2.6**

L39 – "The smooth boundaries and distinct shape of the climatological anticyclone" can be explained further with specific descriptions here. What does 'smooth' mean? Does 'distinct shape' refer to eddy shedding event?

**Response:**

In this case we use "smooth" and "distinct shape" to refer to the clear, smooth, oblong boundary around the time-mean Asian summer monsoon anticyclone as often shown in the literature. This is often shown in isobaric geopotential height (see, e.g., Nützel et al. 2016), isentropic Montgomery streamfunction (see, e.g., Manney et al. (2021)), or potential vorticity (Fig. 14 of our manuscript). We will add some additional context to set the stage for this sentence in our revised submission.

**Comment 2.7**

L65 – A brief mention of why all these species are analyzed together will be useful here.

**Response:**

The rationale for examining these species together is outlined in detail in the preceding paragraph. We will clarify this relationship in the revision.

> **Comment 2.8**
>
> L66 – Here 'further insight' sounds vague. Consider replacing it with more specific terms. For instance, 'analyzing seasonal behaviors or interactions between various processes that have not been analyzed before'.
> L68 – Instead of 'pay attention', mentioning what is new in this work compared to the related work (Tegtmeier et al. and Wright et al.) would be recommended.

**Response:**

We apologize for the misunderstanding. Here we feel that "further insight" is appropriate because what we provide is a new perspective (via the reanalysis-based budget decomposition) on processes that have been extensively analyzed but remain incompletely understood and poorly quantified. More specifically, as outlined in the first two paragraphs of the introduction, many previous studies have examined variability in the composition of the monsoon UTLS from a variety of perspectives, yielding a good qualitative understanding and situational quantitative information on how these processes and covariations contribute. Although the ability of reanalysis products to represent the processes previous studies have shown to be important is an open question, it is important to us to respect the effort and body of work that have established a foundation for us to address that question.

> **Comment 2.9**
>
> L81 – What horizontal grids are used in the gridding?
> L112 – Is the 'replay' technique commonly used or specific to this study?

**Response:**

L81: We adopt the coarsest grid among the evaluated datasets and interpolate all dataset onto this grid for the analysis. This grid corresponds to the Aura MLS Level 3 products on a $2.5° \times 2.5°$ regular latitude–longitude grid.

L112: "Replay" is a technique developed by NASA GMAO. In addition to M2-SCREAM, it has been used in a number of specified dynamics-type model simulations (e.g., Orbe et al. 2017). It is not specific to this study.

> **Comment 2.10**
>
> L140 – The meaning of this sentence is unclear as well. Are specific humidity tendencies produced by parameterized physics and data assimilation?

**Response:**

Specific humidity tendencies ($\partial q/\partial t$) are decomposed into contributions from parameterized physics ($S_{\mathrm{phy}}$), data assimilation ($S_{\mathrm{ana}}$), and moisture flux convergence due to resolved dynamics ($\nabla \cdot (\mathbf{V}q) + \frac{\partial(\omega q)}{\partial p}$). Please see Section 2.2 for details.

> **Comment 2.11**
>
> L176 - What does 'weighted equally' mean here?

**Response:**

In the most common applications of EOF analysis, time variations of two-dimensional (latitude-longitude) spatial patterns are evaluated. In our analysis, we instead flatten the horizontal dimensions and apply the EOF decomposition to vertical-horizontal distributions. One dimension corresponds to both latitude and longitude, while the second corresponds to pressure within the 147 hPa–68 hPa layer. In this context, "weighted equally" means that anomalies in all vertical layers are given the same weight in the decomposition, so that the only weights that apply to the input data are the area weights in the horizontal dimension.

> **Comment 2.12**
>
> L207 – Instead of stating 'no prior expectation', one can try to find if there is a known trend in water vapor.

**Response:**

"No prior expectation" is meant in a roughly Bayesian sense: when developing the analysis, we did not consider whether a linear trend was likely or unlikely to exist in the data. However, as the analysis proceeded, it became clear that the low-frequency principal component exhibited a significant trend, and that the spatial pattern of this trend was distinct from that of interannual variability. Although we are unaware of any previous study highlighting a trend in this region, there have been reports of trends in adjacent regions, such as water vapor in the extratropical lowermost stratosphere (e.g., Dessler et al 2013) or the trend in cold point tropopause temperatures referenced in the discussion (Zolghadrshojaee et al., 2024).
* * *
**Comment 2.13**

Figure 5 – It is striking to see how the left panel (MLS) and the right panel (CAMS) look very different for all the species. It is not easy to understand the explanations of this figure.

**Response:**

This is a key point we wish to emphasize in our manuscript: the observed and reanalysis trends differ, and therefore, caution must be exercised when interpreting these trends. At the suggestion of Dr. Tao (reviewer 1), we have conducted new analysis demonstrating that stratospheric variability associated with PC1 is largely consistent between Aura MLS and the reanalysis products. The differences illustrated in Figure 5 between MLS and reanalysis data thus attributable to different trends in water vapor below the tropopause, in the 147–100 hPa layer. For more details, please refer to our response to comment 1.1 above.
* * *
**Comment 2.14**

L303 – I am not sure how much we can assume that the drift correction for Aura MLS water vapor contributed to the trend. Is it just a speculation or based on some findings?

**Response:**

This an alternative hypothesis, namely: "the drift correction causes Aura MLS to miss a real moistening trend in this region". Because M2-SCREAM includes a positive trend, we have strong confidence that there is a (statistical but not necessarily meaningful) difference in the trend between MLS version 4 and MLS version 5.

We do not assume that this hypothesis is correct, and have largely discounted it because (1) other lines of evidence (like disagreements in the spatial patterns of trends in ozone and CO as remarked in the next sentence) suggest that the alternative hypothesis (i.e., that trends in reanalysis products are unreliable) is more likely and (2) we have a stronger prior expectation that trends in reanalysis products are unreliable. However, it remains a plausible if unlikely alternative hypothesis that thus deserves a place in the discussion. We will make sure that our skepticism about this hypothesis is made clear in the revision.
* * *
**Comment 2.15**

L571 – 'Concurrent variations…' -> This sentence seems to be based on speculations.

**Response:**

Actually not, there are some studies results.

---

## Author Response (AR1)

**Author Reply to Reviewers**

We thank Dr. Mengchu Tao and Reviewer 2 for raising a number of interesting questions. Addressing these questions has already helped tremendously to tighten the analysis and make it more incisive, as described below. The resubmission will be much stronger for your efforts. Initial responses to all general comments and selected technical comments are provided below.

**1 Response to reviewer 1**

**Reviewer 1 evaluations:** This paper focuses on analyzing water vapor variability in the Upper Troposphere–Lower Stratosphere (UTLS) region of the Asian Summer Monsoon (ASM) area. The authors use satellite data from Aura MLS and five atmospheric reanalysis datasets, including MERRA-2, M2-SCREAM, CAMS, ERA5, and JRA-3Q, to conduct a spatiotemporal mode analysis. The main results reveal three key modes of variability, including PC1, large-scale regional water vapor wetting or drying anomalies, PC2 and PC3, intraseasonal oscillations linked to quasi-biweekly variability.

A key conclusion in the paper, from my point of view, is the discrepancies between reanalysis products and Aura MLS data regarding spatial distribution and sign (positive/negative) of water vapor trends. While most reanalysis products show an increasing water vapor trend in the southeast of Asian monsoon during warm season, their spatial characteristics differ significantly from those derived from Aura MLS.

Overall, the paper is novel in design, well-written, and thorough in its data analysis, comparisons of methods, and results. I recommend minor revisions to address the following points for further improvement.

**1.1 General comments**

> **Comment 1.1**
>
> The analyses, particularly those focused on interannual variability and intraseasonal oscillations, are convincing and well-executed. My concerns are mainly about the analysis of PC1 trends. While the discussion clearly identifies the differences in trends between Aura MLS and reanalysis products, I would suggest a deeper exploration of the reasons behind these discrepancies.

**Response:**

Thank you for this suggestion. We have conducted further analysis and incorporated new results accordingly.

As documented by Wright et al., 2025 (see their Figure 1), the cold point tropopause (CPT) during our analysis period is located between 85–95 hPa. In the following, we adopt this estimate of the CPT pressure as the divider between the upper troposphere (UT) and the lower stratosphere (LS). The partial column water vapor (PCWV) above the tropopause is defined as the water vapor integrated from 83 to 68 hPa, whereas PCWV below the tropopause is integrated from 147 to 100 hPa. Figure R1 presents the deseasonalized anomalies of PCWV above the tropopause regressed onto the first principal component (PC1), its trend (PC1$_{\text{TREND}}$), and interannual variability (PC1$_{\text{IAV}}$). Conversely, Figure R2 shows the deseasonalized anomalies of PCWV below the tropopause, regressed onto the PC1 interannual variability and the second principal component (PC2). Here we focus on M2-SCREAM, JRA-3Q, CAMS, and ERA5, excluding MERRA-2 due to its inability to represent variability in the LS.

The spatial distribution of PCWV (Fig. R1) regressed onto PC1 above the tropopause (hereafter, PC1$^{\text{strat}}$) aligns well with the spatial pattern regressed onto PC1 in the UTLS (i.e., Fig. 1 in the manuscript). Meanwhile, the spatial distributions regressed onto PC1 and PC2 below the tropopause (hereafter, PC1$^{\text{trop}}$ and PC2$^{\text{trop}}$, Fig. R2) correspond to anomalies regressed onto UTLS PC2 and PC3, respectively (i.e., Fig. 2 in the manuscript). This clarifies that the PC signals identified for the UTLS PCWV (shown in the manuscript) encompass signals centered in both the LS (represented by PC1 in the manuscript) and the UT (represented by PC2 and PC3 in the manuscript). Specifically, PC1 (in the manuscript) captures interannual variability in the lower stratosphere (PC1$^{\text{strat}}$), above the tropopause, whereas PC2 and PC3 (in the manuscript) reflect subseasonal variability in the upper troposphere (PC1$^{\text{trop}}$ and PC2$^{\text{trop}}$), below the tropopause. The high correlations observed between the UTLS principal components and those above/below the tropopause further support this interpretation (Fig. R3). Consequently, the discrepancies in trend between Aura MLS and reanalysis products predominantly stem from different trends in water vapor in the lower part of the tropopause layer (i.e., 147 hPa to 100 hPa). All reanalysis products simulate a regional moistening trend in water vapor above the tropopause, consistent with Aura MLS (Fig. R1b,e,h,k,n).

[Figure]

Figure R1: Deseasonalized partial-column water vapor (PCWV) anomalies integrated from 83 to 68 hPa regressed onto the (a) first principal component (PC1) from Aura MLS and its (b) trend (PC1$_{\text{TREND}}$) and (c) interannual variability (PC1$_{\text{IAV}}$) components; (d)–(f) same as (a)–(c), but for PC1 from M2-SCREAM; (g)–(i) same as (a)–(c), but for PC1 from JRA-3Q; (j)–(l) same as (a)–(c), but for PC1 from CAMS; (m)–(o) same as (a)–(c), but for PC1 from ERA5; and (p)–(r) the corresponding principal component time series. Principal components (PCs) are based on EOF analysis of vertical and horizontal variations in water vapor for the two Aura MLS pressure levels within 68 hPa–83 hPa, 30°E–130°E, and 15°N–45°N. Red contours mark the location of the Tibetan Plateau, with stippling indicating significance at the 95% confidence level based on Student's $t$ test. The fraction of total variance explained by each mode is listed at the upper right of panels(a,d,g,j,m). Correlations between MLS-based PCs and those based on M2-SCREAM (light red), JRA-3Q (purple), CAMS (light blue), and ERA5 (dark blue) are listed from right to left along the tops of panels (p)–(r).

[Figure]

Figure R2: Deseasonalized partial-column water vapor (PCWV) anomalies integrated from 147 to 100 hPa regressed onto the (a) first principal component (PC1) interannual variability (PC1$_{IAV}$) and (b) the second principal component (PC2) from Aura MLS; (c)–(d) same as (a)–(b), but for PC1$_{IAV}$ and PC2 from M2-SCREAM; (e)–(f) same as (a)–(b), but for PC1$_{IAV}$ and PC2 from JRA-3Q; (g)–(h) same as (a)–(b), but for PC1$_{IAV}$ and PC2 from CAMS; (i)–(j) same as (a)–(b), but for PC1$_{IAV}$ and PC2 from ERA5; and (k)–(l) the corresponding principal component time series. Principal components (PCs) are based on EOF analysis of vertical and horizontal variations in water vapor for the three Aura MLS pressure levels within 100 hPa–147 hPa, 30°E–130°E, and 15°N–45°N. Red contours mark the location of the Tibetan Plateau, with stippling indicating significance at the 95% confidence level based on Student's $t$ test. The fraction of total variance explained by each mode is listed at the upper right of panels(a–j). Correlations between MLS-based PCs and those based on M2-SCREAM (light red), JRA-3Q (purple), CAMS (light blue), and ERA5 (dark blue) are listed from right to left along the tops of panels (k)–(l).

[Figure]

Figure R3: Principal component (PC) time series for the spatial patterns shown in Figs. R1 and R2 based on Aura MLS (black), ERA5 (dark blue), CAMS (light blue), JRA-3Q (purple), and M2-SCREAM (light red) and PCs shown in the manuscript based on Aura MLS (dashed black). Time series are shown for (a) the first principal component (PC1) interannual variability in the UTLS ($PC1_{IAV}$) and above the tropopause ($PC1_{IAV}^{strat}$); (b) the second principal component (PC2) in the UTLS and PC1 interannual variability below the tropopause ($PC1_{IAV}^{trop}$); and (c) the third principal component (PC3) in the UTLS and PC2 below the tropopause ($PC2^{trop}$). PCs in the UTLS are based on EOF analysis of vertical and horizontal variations in water vapor for the five Aura MLS pressure levels within 68 hPa–147 hPa, 30°E–130°E, and 15°N–45°N; PCs above the tropopause are based on EOF analysis of vertical and horizontal variations in water vapor for the three Aura MLS pressure levels within 68 hPa–83 hPa, 30°E–130°E, and 15°N–45°N; and PCs below the tropopause are based on EOF analysis of vertical and horizontal variations in water vapor for the three Aura MLS pressure levels within 100 hPa–147 hPa, 30°E–130°E, and 15°N–45°N. Correlations between MLS-based PCs in the UTLS and those based on MLS above/below the tropopause (black), M2-SCREAM (light red), JRA-3Q (purple), CAMS (light blue), and ERA5 (dark blue) are listed from right to left along the tops of panels (a–c).

[Figure]

Figure R4: Deseasonalized partial-column water vapor (PCWV) anomalies integrated (a) from 147 to 68 hPa and (b) from 147 to 100 hPa regressed onto the first principal component (PC1) trend variability (PC1$_{\text{TREND}}$) from Aura MLS (Fig. R1q); (c)–(d) same as (a)–(b), but for PCWV from M2-SCREAM; (e)–(f) same as (a)–(b), but for PCWV from JRA-3Q; (g)–(h) same as (a)–(b), but for PCWV from CAMS; (i)–(j) same as (a)–(b), but for PCWV from ERA5. Principal components (PCs) are based on EOF analysis of vertical and horizontal variations in water vapor for the two Aura MLS pressure levels within 68 hPa–83 hPa, 30°E–130°E, and 15°N–45°N. Red contours mark the location of the Tibetan Plateau, with stippling indicating significance at the 95% confidence level based on Student's $t$ test.

> **Comment 1.2**
>
> Another question from my side, whether the second reason to doubt the reanalysis-based trends are robust: "trends in cold point tropopause (CPT) temperatures are negative in the southeastern quadrant of the anticyclone where the reanalyses show the largest positive trends in water vapor" (in conclusion). If reanalysis WV increase mainly under the tropopause and thus increase the PCWV, it seems be consistent with OLR/cloud trend (convection increase pattern shown in Figure 6). It thus meets "criteria 3: a plausible physical mechanism". And WV decrease due to local CPT decrease is not a main driver since the mass of WV in the LS is much less than that in the UT. Surely, long-term warming of the tropical cold point tropopause increasing WV outside the monsoon region can be another reason. But I don't see the contradiction between CPT cooling and PCWV increasing over one region.
>
> To shed more lights on this point, my suggestion could be:
>
> 1) the trends (and their spatial characteristics) be further decomposed into contributions above and below the tropopause;
>
> 2) further analysis of the "dyn" and "phy" terms individually (specifically their behaviours above and below the tropopause) in Fig. 7. This could potentially reveal whether convection is the primary driving factor in the trends observed. And this could uncover systematic patterns or consistencies within these two terms from reanalysis datasets.

**Response:**

Thank you for your detailed and insightful comment. We agree that the trend below the tropopause meets the third criterion (but not the second) and have worked to make this clearer in the revised text. This possibility also motivates our call for a more complete analysis of trends in UTLS composition and related fields in this region at the end of section 3.2.

As discussed in our response to Comment 1.1, PC1 in our manuscript primarily corresponds to interannual variability in the humidity of the lower stratosphere (PC1$^{\text{strat}}$, see Fig. R1 and Fig. R3a). Both reanalysis products and observations indicate a regional moistening trend above the tropopause (Fig. R1). Although these moistening trends occur downstream of the positive anomalies in the cold point tropopause (CPT) (Fig. R5a,d,g), consistent with the expectation that a warmer CPT should enhance water vapor content above the cold point, it is difficult to relate these to the enhanced convection in the southeast quadrant. The difficulty arises because there is a trend toward colder cold point temperatures co-located with that increase in convective activity. To evaluate this in more detail, we have examined the changes in tendency terms between these two regions. The results highlight a pair of offsetting trends in the southeastern quadrant: an increase in dynamical moistening (consistent with stronger large-scale ascent; Fig. R6) and a compensating increase in drying due to parameterized physics (consistent with colder CPT temperatures; Fig. R5). These trends in the tendencies are most pronounced in MERRA-2, for which the dynamical term is provided directly.

In an area- and annual-mean sense, trends in resolved transport (DYN; $-\nabla \cdot (\mathbf{V}q) - \partial(\omega q)/\partial p$) are small but positive both above (Fig. R7) and below the tropopause (Fig. R8) in the reanalyses. However, the increases in these transports are offset by a decreasing trend in the physical tendencies ($S_{\text{phy}}$), and sometimes more than offset, as is the case in MERRA-2. The net changes in PHY + DYN trends are inconsistent across different products and are effectively offset by the data assimilation components ($S_{\text{ana}}$), regardless of the altitude relative to the tropopause.

From the above analyses, the increasing trend in 68–83 hPa water vapor in the reanalyses is best explained by an increase of dynamical moistening in the southeast quadrant. Regardless, there are two classes of explanations for the difference between Aura MLS and the reanalysis products:

(1) Aura MLS did not observe an increase because it did not occur in reality or was limited to lower levels (i.e., an error in the reanalyses); or

(2) Aura MLS did not observe an increase because of a sampling bias (e.g., it was limited mainly to a part of the diurnal cycle that MLS does not sample) or some other gap in the observations.

**1.2 Specific comments**

> **Comment 1.3**
>
> Following EQ (2) in Line 160, the terms "Sphy" and "Sana" should be briefly explained. Specifically: What key physical processes related to water vapor are captured by "Sphy" (e.g., condensation, deposition, subsidence, etc.)? What is the role of "Sana," particularly in relation to the data assimilation process? I also wonder whether subgrid-scale mixing is included in the "Sres" term?

**Response:**

Thank you for this suggestion. We have explained these terms in more detail in the revised manuscript. (L166-178)

[Figure]

Figure R5: (a) Changes in cold point tropopause (CPT) temperatures, (b) dynamics term, and (c) physics term based on ERA5 regressed onto the first principal component (PC1) trend variability above the tropopause (PC1$_{\text{TREND}}^{\text{strat}}$) from Aura MLS. (d–f) same as (a–c), but for MERRA-2. (g–i) same as (a–c), but for JRA-3Q. Dynamics and physcial terms are integrated from 147 to 100 hPa. The location of the Tibetan Plateau is marked by a black contour in all panels. Stippling indicates locations where regression slopes are significant at the 95% confidence level based on Student's $t$ test.

[Figure]

Figure R6: Changes in (a) horizontal moisture flux convergence ($-\nabla \cdot (\mathbf{V}q)$), (b) vertical moisture flux convergence ($-\frac{\partial(\omega q)}{\partial p}$) based on ERA5 regressed onto the first principal component (PC1) trend variability above the tropopause (PC1$_{\text{TREND}}^{\text{strat}}$) from Aura MLS. (c–d) same as (a–b), but for JRA-3Q. Horizontal and vertical moisture flux convergence terms are integrated from 147 to 100 hPa. The location of the Tibetan Plateau is marked by a black contour in all panels. Stippling indicates locations where regression slopes are significant at the 95% confidence level based on Student's $t$ test.

[Figure]

Figure R7: Yearly variations in the sum of the dynamics and physics terms (green lines), assimilation increments (yellow lines), dynamics terms (brown lines), physics increments (red lines), and time rate of changes in partial column water vapor (gray boxes) above the monsoon tropopause layer (68-83 hPa) based on (a,b) ERA5, (c,d) JRA-3Q, and (e,f) MERRA-2 over (a,c,e) the southwestern quadrant (15°N–30°N, 30°E–80°E) and (b,d,f) the southeastern quadrant (15°N–30°N, 80°E–130°E) of the monsoon anticyclone. Correlation coefficients between net water vapor tendency time series based on individual reanalyses are listed in the lower left corner of each panel.

[Figure]

Figure R8: Yearly variations in the sum of the dynamics and physics terms (green lines), assimilation increments (yellow lines), dynamics terms (brown lines), physics increments (red lines), and time rate of changes in partial column water vapor (gray boxes) below the monsoon tropopause layer (100–147 hPa) based on (a,b) ERA5, (c,d) JRA-3Q, and (e,f) MERRA-2 over (a,c,e) the southwestern quadrant (15°N–30°N, 30°E–80°E) and (b,d,f) the southeastern quadrant (15°N–30°N, 80°E–130°E) of the monsoon anticyclone. Correlation coefficients between net water vapor tendency time series based on individual reanalyses are listed in the lower left corner of each panel.

$S_{\text{phy}}$ comprises the influences of parameterized physical processes, including cloud microphysics, convection, and turbulent mixing (see Wright et al. (2025), their Fig. 8). Subgrid-scale mixing is included in $S_{\text{phy}}$. $S_{\text{res}}$ is a 'diffusive' residual. We interpret this residual as primarily representing the transport due to high-frequency or small spatial scale motions resolved by the reanalysis model but not by our calculation of $S_{\text{dyn}}$, together with the effects of numerical diffusion. Specifically, we calculate $S_{\text{dyn}}$ on a coarser spatial grid (for ERA5, by a factor 4) and the analysis interval (1 h) is five times longer than the model time step (every 12 minutes). As a result, our calculation is missing a chunk of the transport that the model resolves. Because the model resolves this transport, it does not appear in the physics term. Because our calculation of moisture flux divergence does not resolve this transport, it does not appear in the dynamics term. We have struggled to come up with a good name for this term that clearly conveys what it represents ('resolved mixing' as opposed to 'subgrid-scale mixing'? 'eddy transports'?) and would welcome suggestions. Finally, the residual term also includes the effects of numerical diffusion.

$S_{\text{ana}}$ is the data assimilation term. Although MERRA-2 explicitly provides $S_{\text{ana}}$, this component must be estimated for budgets based on ERA5 and JRA-3Q. We estimate $S_{\text{ana}}$ for ERA5 and JRA-3Q by directly subtracting forecast specific humidities (the model-generated background state before data assimilation) from analysis specific humidities (the final reanalysis product after data assimilation). We then average this difference and multiply by the number of analysis cycles per day (two for ERA5 and four for JRA-3Q) to get an assimilation-related moistening rate in units of per day (alternatively we could divide by the length of the analysis cycle).
* * *
**Comment 1.4**

Figure 1: The titles of the three subplots for MERRA-2 (panels am-cm) are incorrect; they should refer to "PC1" instead of "PC2."
* * *
**Response:**

Thank you for your careful check. However, due to the distinct characteristics of different reanalysis datasets in the upper troposphere and lower stratosphere (UTLS), the order of principal components (PCs) derived from each dataset is not necessarily the same. We therefore attempt to match the modes in terms of their spatial signatures. Under this approach, PC2 in MERRA-2 best corresponds to PC1 in other reanalysis datasets. The reason for this difference is that MERRA-2 relaxes stratospheric water vapor to a zonal-mean climatology with a 3-day relaxation time scale, washing out low-frequency variability in the lower stratosphere. PC1 based on Aura MLS is defined largely by the variance in lower stratospheric water vapor, so MERRA-2 does not represent this mode well.
* * *
**Comment 1.5**

Figure 3 (panel b): Please reduce the y-axis range to between -2 and 2 to allow for better visualization. Consider setting the y = 0 reference line to gray for improved clarity.
* * *
**Response:**

We appreciate your suggestion. The y-axes of all panels in the figure were set to a fixed range of -4 to 4 to facilitate comparison across the panels. However, we acknowledge that this choice makes it more difficult to compare the line profiles in panel b effectively and have revised Figure 3b to address this issue.
* * *
**Comment 1.6**

I wonder why MLS trend show larger positive trend than merra-2 (Fig. 3)? It seems conflict with Fig.1. Does that mean MLS PC1 trend for the whole region is positive? And the positive trend is highly contributable from 35-45N latitude band according to Fig.1 (b)?
* * *
**Response:**

Thank you for raising this question, which highlights a logic error in our presentation of Fig. 1. MERRA-2 exhibits the smallest trend in the low-frequency PC among the datasets we examine (PC2$_{\text{TREND}}$ in MERRA-2). This is primarily because MERRA-2 damps lower stratospheric water vapor variability through relaxation toward a zonal-mean annually-repeating seasonal cycle, resulting in comparatively small year-to-year variability and trend signals. Because the trend in the lower stratosphere (the main point of consistency between Aura MLS and the reanalyses) is small, the low-frequency mode in MERRA-2 is dominated by variations below the tropopause and tilts more toward PC2$_{\text{IAV}}$. Conversely, MERRA-2 shows larger PCWV anomalies when regressed onto the trend component compared to MLS. This is because all regressions were shown for PC$_{\text{TREND}}$ = 1. This approach is appropriate when the principal components are standardized, as in Fig. 2 and column (a) of Fig. 1. However, because we have not re-standardized PC1$_{\text{TREND}}$ and PC1$_{\text{IAV}}$, the magnitude of the water

vapor anomalies associated with both modes is overstated in columns (b) and (c) of Fig. 1. This overstated amplitude is more pronounced for MERRA-2, because time variance in $PC2_{TREND}$ based on MERRA-2 is smaller than that for $PC1_{TREND}$ based on the other datasets. To correct this, we kept the current time series of the trend and IAV parts of the low-frequency principal component but show regressions in columns (b) and (c) of Fig. 1 for the re-scaled trend and IAV time series.

Regarding your second question, yes, the positive trend in the full 147–68 hPa partial column is most robust along the northern edge of our analysis domain. Figs. R1 and R4 provide further insight into these spatial patterns, as discussed in our response to comment 1.1 above.

**2 Response to reviewer 2**

**Summary evaluation:** This work focuses on three leading modes of interannual variability in water vapor in the tropopause layer using the measurements from the MLS satellite and multiple reanalysis products. The first mode is linear trend and interannual variability in regional-scale anomalies, which show some differences in the satellite data and reanalysis. The second and third modes are related to anomalies within the monsoon anticyclone, such as, variabilities within the quadrants and a horizontal east-to-west dipole structure. The results show reanalysis captures the modes of variability in water vapor in the upper troposphere and lower stratosphere and the physical processes controlling them. The results shown here are based on comprehensive, thorough and very detailed analyses. However, I did not see clear motivation and goal of the work. Below are my comments for the authors might take into consideration.

**2.1 General comments**

> **Comment 2.1**
>
> L1 (Abstract): Rather than starting with 'we describe', I recommend start with some background information including why the Asian summer monsoon is important and what the goal of this study is. This will make the abstract more appealing.

**Response:**

Thank you for this suggestion. We removed this background information from the abstract at submission due to word count requirements. In the revised manuscript, we have worked to formulate a more appealing abstract that also meets the 250-word length requirement.

**Revised abstract:** "The upper-level anticyclone above the Asian summer monsoon (ASM) greatly influences variations in stratospheric water vapor, which in turn have significant effects on climate. An impending data gap underscores the need to evaluate the reliability of recent reanalysis products in this region. Here, we describe three leading modes of deseasonalized water vapor variability in the tropopause layer (147-68 hPa) above the ASM. The first mode describes regional-scale moist or dry anomalies that peak in the lower stratosphere on interannual scales. Separating this mode into linear trend and detrended components, we find that the spatial pattern and sign of the trend disagree between observations and reanalyses. These discrepancies arise from different responses in the upper troposphere despite broad agreement in the lower stratosphere. Regional water vapor budgets suggest that the stratospheric trend originates outside the monsoon region, beyond our analysis domain. Interannual variability is more consistent, and arises mainly from the pre-monsoon influence of the quasi-biennial oscillation. The second mode features anomalies arcing around the northern flank of the anticyclone with weaker opposing anomalies in the southeast, while the third mode features a horizontal dipole oriented east-to-west. These two modes often vary in quadrature as quasi-biweekly waves propagate across the region, but also vary independently when other modes of convective variability manifest in similar areas. Despite lingering questions on the linear trend, mean biases, and data assimilation effects, the consistency between observation- and reanalysis-derived variability demonstrates that reanalyses are increasingly able to capture the processes controlling water vapor near the tropopause."

> **Comment 2.2**
>
> L16 (Introduction): It was mentioned that the goal of this study is to provide further insight into the mechanisms governing variations in water vapor. More specific information could be added here. What are examples of the mechanisms that we need to understand further? What variations in water vapor is discussed here? What is the science goal? A clear motivation and some scientific context of this work will be necessary. This work maybe relevant to the fact that there will be less observations of stratospheric water vapor available from satellites in the near future.

**Response:**

Thank you for raising these questions, which provide guidance for us to present the work in a way that will be clearer and easier for readers to understand.

The research gaps and processes that we target in this work are articulated at the end of the first paragraph in the introduction: "Despite much progress in recent years, uncertainties remain regarding the relative influences and interplays among convective transport, the large-scale circulation, and thermodynamic structure near the tropopause in controlling humidity and composition in the upper troposphere and lower stratosphere (UTLS) above the ASM." We revisit this in the closing paragraph of the introduction, in the sentence following the one you reference: "We pay particular attention to characteristic patterns of covariability among convective activity, circulation patterns, and trace gas concentrations [as represented in current reanalysis

systems]". To this, we have added that we evaluate the roles of data assimilation relative to those of parameterized physics and large-scale dynamics in allowing reanalysis systems to reproduce recurrent patterns of water vapor variability in the UTLS above the ASM. (L77-79)
* * *
**Comment 2.3**

In depth and very detailed analyses and descriptions of the results are presented in this work. I found it rather hard to understand all the detailed descriptions of figures. Many of the sentences are long and the description of results contains some speculation, besides facts. It would be helpful if some of the long sentences are split into multiple short sentences and simplify the descriptions.

**Response:**

We appreciate your suggestion and have made changes to the text accordingly (see track-changes version).
* * *
**Comment 2.4**

It would also be helpful to include some context of the results from this work relative to previous studies throughout the main text. How is the result shown here different from previous work? Are they consistent with or different from previous work? This will help understand the results more scientifically. Is EOF analyses giving us new information that has not been discovered?

**Response:**

Thank you for this suggestion. Although direct comparisons are not always possible due to methodological differences, we have provided a discussion of our results in the context of previous work in section 4. It is also worth noting that the EOF analysis is primarily an exploratory tool. The first question we address with this tool is: are recurrent patterns similar between high-quality observations and reanalysis products? Having established that they are largely similar and delineated areas of disagreement, we address a pair of follow-up questions: do these recurrent patterns derive from clear physical and dynamical mechanisms? If so, do the reanalyses represent these mechanisms consistently?

The decision to use EOF decomposition in this work was motivated by both the widespread use of this procedure in climate science and the platform it provides for us to identify and analyze recurrent patterns of water vapor anomalies. The EOF approach has known limitations (susceptible to non-stationarity, unable to cleanly identify propagating modes) and many other statistical tools could provide a similar platform. Moreover, EOF patterns in isolation can be difficult to interpret and are often not physically meaningful. Although the use of vertically resolved information in identifying these patterns has not to our knowledge been applied to this region before, it is not the EOF analysis itself that provides new information, but rather our detailed decomposition of how the reanalyses generate the recurrent anomaly patterns identified by the EOF decomposition.
* * *
**Comment 2.5**

I think it would be helpful to provide some outlook. For instance, information about which reanalysis products represent dynamical or thermodynamical processes near the monsoon region well so that we can trust?

**Response:**

Thank you for this suggestion. We have collected and collated this information from the manuscript into a summary in the final section. (L741-757)

**2.2 Specific Comments**

**Comment 2.6**

L39 – "The smooth boundaries and distinct shape of the climatological anticyclone" can be explained further with specific descriptions here. What does 'smooth' mean? Does 'distinct shape' refer to eddy shedding event?

**Response:**

In this case we use "smooth" and "distinct shape" to refer to the distinct oblong boundary around the time-mean Asian summer monsoon anticyclone as often shown in the literature. This is most commonly plotted in isobaric geopotential height

(see, e.g., Nützel et al. 2016), isentropic Montgomery streamfunction (see, e.g., Manney et al. 2021), or potential vorticity (Fig. 14 of our manuscript). We have added some additional context before this sentence in our revised submission. (L39–42)
* * *
**Comment 2.7**

L44 – 'constrained' could be replaced with 'understood'.
L51 – Here 'associations' sound vague. I am wondering this could be replaced with some other word.
L57 – Seasonal dilution of ozone in the UTLS above the monsoon anticyclone...
L60 – air to the -> air into the
* * *
**Response:**

L44: Changed as suggested. (L47)

L51: We revised the sentence to: "Variations in ozone and carbon monoxide (CO) in the UTLS are strongly correlated with those in water vapor." (L54)

L57: Thank you for your careful check. We have revised the text to "Seasonal dilution of ozone in the UTLS above the monsoon ...". We have elected not to write "above the monsoon anticyclone" because the anomalies we reference are within the monsoon anticyclone (which caps the monsoon) rather than above it. (L59)

L60: Changed as suggested. (L63)
* * *
**Comment 2.8**

L65 – A brief mention of why all these species are analyzed together will be useful here.
* * *
**Response:**

The rationale for examining these species together is outlined in detail in the preceding paragraph (L54-65). We have clarified this relationship in the revision. (L68-69)
* * *
**Comment 2.9**

L66 – Here 'further insight' sounds vague. Consider replacing it with more specific terms. For instance, 'analyzing seasonal behaviors or interactions between various processes that have not been analyzed before'.
* * *
**Response:**

We apologize for the misunderstanding. Here we feel that "further insight" is appropriate because what we provide is a new perspective (via the reanalysis-based budget decomposition) on processes that have been repeatedly analyzed but remain incompletely understood and poorly quantified. More specifically, as outlined in the first two paragraphs of the introduction, many previous studies have examined variability in the composition of the monsoon UTLS from a variety of perspectives, yielding a good qualitative understanding and situational quantitative information on how these processes and covariations contribute. Although the ability of reanalysis products to represent the processes previous studies have identified as important remains an open question, it is important to us to respect the effort and body of work that have established a foundation for us to address that question.
* * *
**Comment 2.10**

L68 – Instead of 'pay attention', mentioning what is new in this work compared to the related work (Tegtmeier et al. and Wright et al.) would be recommended.
* * *
**Response:**

We have added a sentence "Whereas those studies focused mainly on climatological characteristics, this work focuses on variability beyond the seasonal cycle." (L77-79)

**Comment 2.11**

L77 - A couple examples of the reanalysis products included in the supplement material would be helpful.

**Response:**

Examples providing intercomparison of individual reanalyses are provided in Figs. 1–3, Figs. 6–8. Fig. 11, and Fig. 16. Additional results showing multiple reanalyses are left to the supplement to conserve space.

**Comment 2.12**

L81 – What horizontal grids are used in the gridding?

**Response:**

We interpolate all datasets onto the coarsest grid among the evaluated datasets for the analysis. This grid corresponds to the Aura MLS Level 3 products on a 2.5°×2.5° regular latitude–longitude grid. (L89)

**Comment 2.13**

L100 – Does this sentence mean that the metrics are used to distinguish the effects of convection, thermodynamic conditions and transport?

**Response:**

Yes, this sentence means that the metrics described in the following sentences are used to evaluate the effects of deep convection (OLR, non-radiative diabatic heating) and thermodynamic constraints on water vapor (CPT).

**Comment 2.14**

L112 – Is the 'replay' technique commonly used or specific to this study?

**Response:**

"Replay" is a technique developed by NASA GMAO. In addition to M2-SCREAM, it has been used in a number of specified dynamics-type model simulations (e.g., Orbe et al. 2017). It is not specific to this study.

**Comment 2.15**

L121 – I find it hard to understand this sentence.

**Response:**

Thanks for your comment. Humidity in current meteorological reanalyses is constrained by satellite radiances and direct measurements with limited spatial coverage (radiosondes, aircraft, etc). Direct constraints on UTLS water vapor are weak because water vapor amounts are small at these altitudes and are thus difficult to retrieve. Radiosonde measurements have strict altitude or relative humidity cut-offs and are typically not used under the conditions that prevail in the ASM UTLS. As a result, the weak tails of deep vertical weighting functions can lead to assimilation increments at lower altitudes making a correlated 'imprint' on water vapor in the UTLS, which causes problems if model biases in the LS are not correlated with those at lower altitudes. We have revised this sentence to: "In addition, the deep vertical weighting functions of some assimilated satellite radiances may lead to data assimilation targeted at the troposphere producing correlated increments at and above the tropopause." (L128-129)

**Comment 2.16**

L140 – The meaning of this sentence is unclear as well. Are specific humidity tendencies produced by parameterized physics and data assimilation?

**Response:**

Specific humidity tendencies ($\partial q/\partial t$) are decomposed into contributions from parameterized physics ($S_{\mathrm{phy}}$), data assimilation ($S_{\mathrm{ana}}$), and moisture flux convergence due to resolved dynamics ($\nabla \cdot (\mathbf{V}q) + \frac{\partial(\omega q)}{\partial p}$). Please see Section 2.2 for details.
* * *
**Comment 2.17**

L159 – Here 'mechanisms behind' could be replaced with a sentence with more specific terms. For instance, 'contributions from physical processes in determining temporal variability of water vapor'.

**Response:**

We replaced the sentence with more specific terms: "The vertically-resolved water vapor budget is used to diagnose the contributions of parameterized physical processes, resolved dynamical transport, and data assimilation toward determining the spatio-temporal variability of UTLS water vapor in the meteorological reanalyses." (L166-168)
* * *
**Comment 2.18**

L171 (paragraph) – This paragraph contains many steps. It is hard to follow each step based on the writing. It would be easier if there is a simple diagram showing the flow.

**Response:**

Thank you for your comment. Because the analysis procedure outlined in this paragraph is fairly common and several of the figures in our manuscript illustrate these steps, we prefer not to incorporate an additional schematic diagram. However, to make the paragraph easier to read, we have added a paragraph break before at "Separate EOF analyses...".
* * *
**Comment 2.19**

L176 - What does 'weighted equally' mean here?

**Response:**

In the most common applications of EOF analysis, time variations of two-dimensional (latitude-longitude) spatial patterns are evaluated. In our analysis, we instead flatten the horizontal dimensions and apply the EOF decomposition to vertical-horizontal distributions. One dimension corresponds to both latitude and longitude, while the second corresponds to pressure within the 147 hPa–68 hPa layer. In this context, "weighted equally" means that anomalies in all vertical layers are given the same weight in the decomposition, so that the only weights that apply to the input data are the area weights in the horizontal dimension. (L186-187)
* * *
**Comment 2.20**

L181 – What does 'reordered for consistency' mean?

**Response:**

Different reanalysis datasets produce distinct eigenmodes that are not necessarily arranged in the same order. For example, PC2 based on MERRA-2 best aligns with PC1 based on the other datasets, while PC4 of M2-SCREAM aligns with PC3 of MLS, ERA5, CAMS, JRA-3Q, MERRA-2. We re-order the principal components to maximize consistency with the spatial patterns based on the Aura MLS results. This has been done first subjectively and then confirmed by computing pattern correlations. Please also refer to our response to comment 1.4 above.
* * *
**Comment 2.21**

L184 – 'underlying mechanisms' can be further explained here.

**Response:**

The underlying mechanisms are discussed in the following section (Section 3). We prefer to keep these detailed explanations separate from our description of the methodology.

**Comment 2.22**

L207 – Instead of stating 'no prior expectation', one can try to find if there is a known trend in water vapor.

**Response:**

"No prior expectation" is meant in a roughly Bayesian sense: when developing the analysis, we did not consider whether a linear trend was likely or unlikely to exist in the data. However, as the analysis proceeded, it became clear that the low-frequency principal component exhibited a significant trend, and that the spatial pattern of this trend was distinct from that of interannual variability. Although we are unaware of any previous study highlighting a trend in this region, there have been reports of trends in adjacent regions, such as water vapor in the extratropical lowermost stratosphere (e.g., Dessler et al 2013) or the trend in cold point tropopause temperatures referenced in the discussion (Zolghadrshojaee et al., 2024).

**Comment 2.23**

L218 – Can be more specific in 'indirect indication' here.

**Response:**

This is an 'indirect indication' because M2-SCREAM assimilated MLS v4. M2-SCREAM assimilates only Aura MLS retrievals, so its representation of water vapor in the UTLS will match variability observed by Aura MLS v4 closely but not exactly.

**Comment 2.24**

L239 – I wonder what is causing 'moistening and drying' in the anticyclone?
L245 – Also what processes contribute to this dipole structure?

**Response:**

L239, L245: The processes that contribute to the moistening and drying and the dipole structure are described in Section 3.3 (now 3.4). The current section provides an overview of the principal components (PCs). To clarify this, we have added a new subsection and indicated "This section provides brief descriptions of the identified modes, which are analyzed in detail in the following sections." (L201) at the beginning of the new subsection.

**Comment 2.25**

L267 & L274 – PC2 & PC3: Are these results consistent with what Randel et al. (2015) have shown?

**Response:**

Thanks for this question. A comparison of our results with those of Randel et al. (2015) is provided in the discussion (L631-637 of the original manuscript; L657-663 of the revised submission). More generally, the composite OLR anomalies reported by Randel et al., (2015, their Fig. 5) are similar to the OLR anomalies associated with PC2 in our results (Fig. 13a). Their composite horizontal temperature anomalies (their Fig. 8) are also consistent with MSF in our PC2 (Fig. 14b) and their vertical temperature anomalies (their Fig. 9) align with our temperature distribution in PC2 (Fig. 14a). Although no results from Randel et al. (2015) match directly with our PC3, we note that methodological differences make a one-to-one comparison difficult. In particular, Randel et al. (2015) used lag relationships in their work. In this light, the quadratic evolution of our PC2 and PC3 broadly fits within their results, and the differences indicate the different perspectives provided by these two distinct approaches.

**Comment 2.26**

Figure 5 – It is striking to see how the left panel (MLS) and the right panel (CAMS) look very different for all the species. It is not easy to understand the explanations of this figure.

**Response:**

This is a key point we wish to emphasize in our manuscript: the observed and reanalysis trends differ, and therefore, caution must be exercised when interpreting these trends. At the suggestion of Dr. Tao (reviewer 1), we have conducted

new analysis demonstrating further that stratospheric variability associated with PC1 is largely consistent between Aura MLS and the reanalysis products. The differences illustrated in Figure 5 between MLS and reanalysis data are thus attributable to different trends in water vapor below the tropopause in the 147–100 hPa layer. For more details, please refer to our response to comment 1.1 above.
* * *
**Comment 2.27**

L303 – I am not sure how much we can assume that the drift correction for Aura MLS water vapor contributed to the trend. Is it just a speculation or based on some findings?

**Response:**

This is an alternative hypothesis, which could be explicitly articulated as: "the drift correction causes Aura MLS to miss a real moistening trend in this region". Because M2-SCREAM includes a positive trend, we have strong confidence that there is a (statistical but not necessarily meaningful) difference in the trend between MLS version 4 and MLS version 5.

We do not assume that this hypothesis is correct, and have largely discounted it because (1) other lines of evidence (such as disagreements in the spatial patterns of trends in ozone and CO as remarked in the next sentence) suggest that the alternative hypothesis — that trends in reanalysis water vapor near the tropopause are unreliable — is more likely and (2) we have a stronger prior expectation that trends in reanalysis products are unreliable. However, it remains a plausible if unlikely alternative explanation and thus deserves a place in the discussion.
* * *
**Comment 2.28**

L307 – 'mechanisms behind' sounds vague. This sentence could simply be 'it is important to understand the reanalysis-based trend'.

**Response:**

We have revised this sentence to state: "Figure 6 addresses the third criterion, as it is important to understand the reanalysis-based trend even if this trend is spurious" (L321)
* * *
**Comment 2.29**

L330 – 'The robust. . .southeast.' could be split in two sentences.

**Response:**

We have split this sentence into two: "The robust nature of these CPT temperature changes is consistent with increased convection and cloud cover over the Tibetan Plateau and areas to its southeast. Although CPT tenmperatures are influenced by model physics and vertical resolution, they are also well constrained by observational data assimilation (Tegtmeier et al. 2020)." (L348-351)
* * *
**Comment 2.30**

L340 – 'largest increasing trends' -> Is this trend based on reanalysis?

**Response:**

Yes, this is based on reanalysis. We have added new analysis and deleted that sentence. (L332-338)
* * *
**Comment 2.31**

L363 – Need citations for this sentence. Also please include the sources for the ONI and the QBO time series.

**Response:**

L363: We have added references regarding the effects of QBO and ENSO on water vapor in the monsoon tropopause layer (L388-389). The sources of the ONI and QBO time series are provided in the 'Data Availability' section: The Berlin QBO time series has been acquired from the updated archive maintained by the Karlsruhe Institute of Technology. The Oceanic Niño Index time series was acquired from the National Oceanic and Atmospheric Administration (NOAA) Climate

Prediction Center (https://www.cpc.ncep.noaa.gov/data/indices/) index calculated with centered base periods using version 5 of the Extended Reconstructed Sea Surface Temperature.
* * *
**Comment 2.32**

P22, Figure 10 – This is a busy figure. I am wondering if this figure could be split into two, or the longitude range could be reduced to cover only the monsoon area.
* * *
**Response:**

Figure 10: The region depicted in this figure is consistent with those in the previous figures. We prefer to prioritize this consistency and have therefore chosen not to simplify the figure. The figure is divided into columns each representing a single reanalysis and rows each representing one term in the tendency budget.
* * *
**Comment 2.33**

L507 – The paragraph starting with 'Oscillations of the jet. . . ' is complicated. I would suggest read it again and revise it if possible.
* * *
**Response:**

Although the paragraph beginning with "Oscillations of the jet..." includes two lengthy sentences, we believe the content to be clear. To enhance readability, we have revised the text as follows: "Oscillations of the jet stream due to the restoring force on the poleward shift upstream of the Tibetan Plateau produce a related anomaly downstream over northern China. Each center spans approximately 20–25° longitude, separated by most of the width of the Tibetan Plateau (also ∼25° longitude). These distances indicate a zonal wavenumber of 8–9. With a background wind speed of 25–30 m s$^{-1}$, the anomaly over northern China should appear about five days after the forcing anomaly west of the Tibetan Plateau, in line with the timescales shown in Fig. 12. Convection over the Indus River valley remains intense into PC2+ (Fig. 14a). The tail of enhanced high cloud cover and reduced OLR is suggestive of the 'up-and-over' transport mechanism that often links lowland monsoon depressions to episodes of strong summer precipitation over the southwestern Tibetan Plateau (Dong et al., 2016,2017)." (L533-540)
* * *
**Comment 2.34**

L571 – 'Concurrent variations. . . ' -> This sentence seems to be based on speculations.
* * *
**Response:**

There is an element of speculation, yes, but it is an informed speculation based on what we know of the physics. Both ideas are based on the same feature: the active core is anomalously cold. First, because colder temperatures produce an upward distortion of the isentropic surface, isentropic transport toward the center of the active core will be more upward in pressure coordinates. Second, colder temperatures reduce the ambient temperature $T$ relative to the radiative equilibrium temperature $T_{\mathrm{eq}}$. Adopting the Newtonian cooling approximation (e.g., Fueglistaler et al. 2009):

$$\frac{Q_{\mathrm{rad}}}{c_p} \approx -\alpha \left(T - T_{\mathrm{eq}}\right) \tag{1}$$

we can therefore expect heating rates to be more positive and diabatic mass fluxes more upward in potential temperature coordinates. Both expectations are borne out by regressing pressure vertical velocities ($\omega$) and net radiative heating ($Q_{\mathrm{rad}}$) onto PC3.
* * *
**Comment 2.35**

L580 – 'which is. . . local dehydration. . . vortex' -> I am not sure what this means. Does this mean condensation due to cold temperature?
* * *
**Response:**

L580: Yes, the anomalously cold temperatures in the active core reduce saturation values causing excess water vapor to condense. It is often assumed that most if not all of this condensate falls out of the air mass, thereby reducing the total water mixing ratio. A long line of studies have established that the coldest temperatures encountered during transit play a dominant

role in determining how much water vapor enters the stratosphere (Mote et al., 1996; Randel et al., 2006; Liu et al., 2010; Schoeberl et al., 2012). Co-location of convective sources with cold temperatures thus constrains the source of water vapor to the UTLS.
* * *
**Comment 2.36**

L582 – 'highly coherent' could be replaced with 'coherent'. What does 'dynamical reorganization' mean?

**Response:**

L582: Changed as suggested. (L608)

Here, 'dynamical reorganization' refers to coherent variations of water vapor, temperature, radiative heating, MSF, and other trace gases, which are all interconnected. In many cases, changes in one variable imply changes in others, sometimes causally and other times correlatively. For example, a weakening of convection produces a reduction of MSF, indicating a reduced tropospheric thickness. This change is associated with a subsidence of lower stratospheric air, producing positive anomalies in temperature and ozone near the tropopause. Warm anomalies imply reduced radiative heating (stronger longwave emission), while geostrophic adjustment produces anomalous winds that alter water vapor transport within the anticyclone. We believe that 'dynamical reorganization' neatly summarizes this cascade of processes linking variations in water vapor to covariations in dynamical and thermodynamic fields but we are open to suggestions.
* * *
**Comment 2.37**

L611 (Discussion) – I would like to suggest adding a few sentences at the beginning explaining why EOF analyses were used in this work. Comparing with previous work which did not use EOF analyses, and their limits would be useful.

**Response:**

Thank you for this suggestion. Here, EOF is merely a statistical technique that identifies the principal components, similar to composites and other statistical methods. These principal components represent the characteristic patterns of interannual and subseasonal variability in water vapor. We employ EOF because of its convenience and widespread use (see our response to comment 2.4 for further details). Given that most readers will already be familiar with this technique and considering the length of the paper, we believe additional explanations are unnecessary.
* * *
**Comment 2.38**

L627 – Can 'mechanisms behind' be just 'role of'?

**Response:**

We don't think that "role of" works in this instance because this paragraph is about the mechanisms involved in establishing covariability among convection, composition, and circulation. We have changed "mechanisms behind ..." to "mechanisms that link convection, composition, and circulation variability" (L653)
* * *
**Comment 2.39**

L682 – 'mechanisms behind' could be just 'physical processes?

**Response:**

L682: Thank you for your comment. Because our analysis also evaluates the dynamical processes involved, we have changed the phrasing from "mechanisms behind" to "physical and dynamical processes behind" (L707)
* * *
**Comment 2.40**

L684 – 'The results serve. . . ' -> Here, I would like to know what we have learned from this paper not what was done in this paper.

**Response:**

Here we presume that you are referring to the previous two sentences. Thank you for raising this point, as we did not use this prime position well in the original manuscript. In the revision, we have replaced the leading two sentences of the conclusions with "In this paper, we have shown that current reanalysis products are largely able to reproduce the leading modes of variability in tropopause layer water vapor above the Asian summer monsoon. Using a range of dynamical and thermodynamic fields, we have further investigated the physical and dynamical processes behind these modes of variability as represented in the reanalysis products."

The final sentence of that paragraph is intended as a lede to the closing paragraphs that follow, which summarize what we have learned in more detail: our assessment of the consistency across reanalyses, their points of agreement and disagreement, and the processes and mechanisms that play central roles in their representations of water vapor variability in this region. Our primary targets in this work are reanalysis intercomparison and assessment, in line with our submission of the work to a special issue dedicated to reanalysis intercomparison.